# Colony demographics shape nest construction in *Camponotus fellah* ants

**Harikrishnan Rajendran†, Roi Weinberger, Ehud Fonio, Ofer Feinerman\***

Department of Physics of Complex Systems, Weizmann Institute of Science, Rehovot, Israel

**\*For correspondence:**
ofer.feinerman@weizmann.ac.il

**Present address:** †Institute of Science and Technology Austria, Klosterneuburg, Austria

**Competing interest:** The authors declare that no competing interests exist.

## eLife Assessment

This study presents an **important** finding that ant nest structure and digging behavior depend on ant age demographics for a ground-dwelling ant species (Camponotus fellah). By asking whether ants employ age-polyethism in excavation, the authors address a long-standing question about how individuals in collectives determine the overall state of the task they must perform. The experimental evidence that the age of the ants and the group composition affect the digging of tunnels is **convincing**, and their model is able to replicate the colony's excavation dynamics qualitatively, results that may prove to be a key consideration for interpreting results from other studies in the field of social insect behavior.

**Abstract** The ant nest serves as the skeleton of the ant superorganism. Similar to a skeleton, the nest expands as the colony grows and requires repair after catastrophic events. We experimentally compared nest excavation in colonies seeded from a single mated queen and allowed to grow for 6 months to excavation triggered by a catastrophic event in colonies with fixed demographics, where the age of each worker, including the queen, is known. The areas excavated by equal group sizes differed significantly between these conditions: heterogeneous populations in naturally growing colonies as well as cohorts of young ants dig larger areas than old ant cohorts. Moreover, we find that younger ants tend to dig slanted tunnels while older ants dig straight down. This is a novel form of age polyethism, where an ant's age dictates not only her likelihood to engage in a task but also the way she performs the task. We further present a quantitative model that predicts that under normal growth, digging is predominantly performed by the younger ants, while after a catastrophe, all ants dig to restore lost nest volume. The fact that the nests of naturally growing colonies exhibit slanted tunnels strengthens this prediction. Finally, our results indicate how a colony's demographic and physical history are sketched into the current structure of its nest.

## Introduction

Nests occur across a range of biological scales and serve various functions such as providing refuge, storage, transport, mating, and micro-climate regulation (*Sudd, 1969*; *Hölldobler and Wilson, 1990*; *Seeley and Morse, 1976*; *Winston, 1987*; *Tschinkel, 1987*; *Seeley and Morse, 1976*; *Pratt, 2004*; *O'Fallon et al., 2023*; *Spezie and Fusani, 2023*; *Magalhaes et al., 2017*; *Pinter-Wollman, 2015*). Among the diversity of nests, those constructed by social insects are highly complex in terms of function, architecture, and construction mechanisms. These nests are created through digging or modifying a substrate and can be arboreal or underground (*Tschinkel, 2003*; *Tschinkel, 2004*; *Tschinkel, 2005*); (*Blum, 1969*; *Bollazzi and Roces, 2007*; *Bollazzi et al., 2008*; *Bollazzi and Roces, 2010*; *Avinery et al., 2023*). Previous studies have demonstrated that these structures emerge from the interaction between individual workers and the environment in a decentralized manner, showcasing

an exemplary instance of biological self-organization (*Bonabeau et al., 1997*; *Sneyd et al., 2001*; *Cassill et al., 2002*; *Deneubourg and Franks, 1995*; *Theraulaz et al., 1998*; *Buhl et al., 2005*; *Toffin et al., 2009*; *Franks and Deneubourg, 1997*; *Gravish et al., 2012*; *Pinter-Wollman et al., 2018*, *Traniello and Rosengaus, 1997*).

Previous studies on nest construction and dynamics often addressed mature colonies of fixed size and over short time scales (*Rasse and Deneubourg, 2001*; *Buhl et al., 2005*; *Buhl et al., 2004*; *Kwapich et al., 2018*). They show that for ant groups of fixed size, nest volume grows logistically to a saturation volume that scales linearly with population size (*Tschinkel et al., 2015*; *Tschinkel, 1999*; *Mikheyev and Tschinkel, 2004*; *Rasse and Deneubourg, 2001*; *Buhl et al., 2005*; *Buhl et al., 2004*). This led to the hypothesis that workers can perceive the density of ants within the nest and adjust their digging activity accordingly. This mechanism may hold in relatively small populations and simple architectural designs. However, for more complex structures where ants congregate in specific chambers, workers are less likely to assess the overall nest density (*Rasse and Deneubourg, 2001*; *Buhl et al., 2005*; *Buhl et al., 2004*). In such cases, nest excavation is enhanced by recruiting additional workers from the other areas of the nest or facilitated by humidity or pheromones (*Aguilar et al., 2018*; *Gordon, 2010*; *Burkhardt, 1998*; *Perna and Theraulaz, 2017*; *Rasse and Deneubourg, 2001*; *Buhl et al., 2005*; *Deneubourg and Franks, 1995*; *Gravish et al., 2012*).

Ant colonies are composed of ants that vary in age. It is well known that the task allocation of individual workers can change throughout their lifetimes (age polyethism) (*Gordon, 1989*; *Gordon, 2016*; *Pinter-Wollman et al., 2012*; *Robinson, 1992*; *Seid and Traniello, 2006*; *Sendova-franks and Franks, 1995*; *Mersch et al., 2013*; *Beshers and Fewell, 2001*; *Dornhaus, 2008*; *Gautrais et al., 2002*). However, it is unknown which of the work groups in a developing colony participate in the digging effort.

Additionally, ant colonies are comprised of the reproductive caste and one or more worker castes that can divide labor among themselves (*Gordon, 1989*; *Sendova-franks and Franks, 1995*; *Theraulaz et al., 1998*; *Mersch et al., 2013*; *Ulrich et al., 2018*; *Jeanson et al., 2007*; *Fewell and Page, 1999*). Thus, colony demographics play a significant role in the development of nest structure, and its role needs to be considered.

One way to view the social insect nest is from a colony-superorganism perspective (*Hölldobler and Wilson, 1990*; *Bonabeau et al., 1997*; *Sneyd et al., 2001*; *Deneubourg and Franks, 1995*; *Franks and Deneubourg, 1997*). Within this analogy, the social insect nest can be likened to the superorganism's skeleton. Inspired by the processes of skeletal growth and bone healing after fracture, we set out to find how the volume of an ant nest is regulated during ordinary colony development and following a catastrophic collapse. Specifically, we were interested in the following questions: (a) How does a colony regulate the co-expansion of the population and the nest as the colony develops from a single queen to tens of workers? (b) How does a colony recover nest volume following a catastrophic collapse? (c) Which ants participate in nest excavation and does this answer differ between the cases (a) and (b)?

We performed two sets of experiments to study how a *Camponotus fellah* ant colony regulates the volume of its nest. *C. fellah* ants are native to the Near East and North Africa, particularly found in countries like Israel, Egypt, and surrounding arid and semi-arid regions, where they prefer to nest in moist, decaying wood, including tree trunks, branches, or stumps (*Wilson, 2003*; *Vonshak and Shlagman, 2009*). These ants live in monogynous colonies, ranging in size from a few dozen to several thousand individuals. Their nests are commonly built in a sand-loamy mix—a combination of sand, soil, clay, or gravel—providing structural stability and moisture retention (*Mersch et al., 2017*). They are typically found under rocks, in the crevices of dried vegetation, or within dry, sandy soils, sometimes in areas with loose gravel (Materials and methods).

The '*Colony-Maturation*' experiment observed the development of colonies for up to 6 months, starting from a single fertile queen and progressing to colonies with established worker populations. We used these experiments to track the coexpansion of the colony and its nest for an extended period (up to over 6 months). In another set of experiments, the '*Fixed-Demographics*' experiments, we monitored excavation by an ant group of determined size and age, which included the ant queen. We used these experiments to study the nest volume recovery following catastrophic events and the role of different age groups in the digging process.

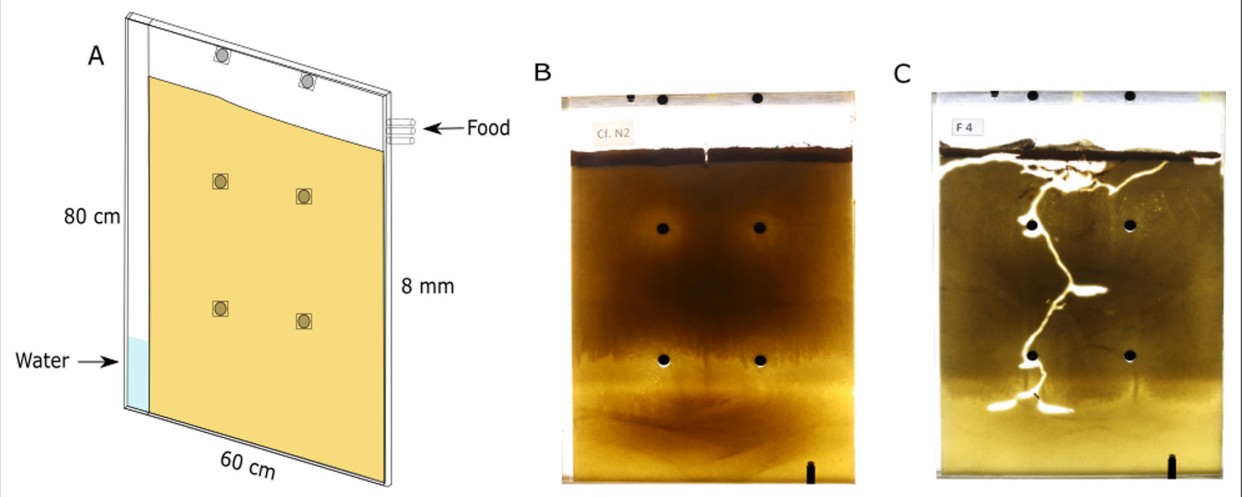

**Figure 1.** Experimental system and nest architecture. (**A**) We used large nest frames 80 × 60 × 0.8 cm³ to allow for long-term excavation. Nest frames are filled with homogeneous fine sand and wet regularly for structural cohesiveness. Water infiltrates through the water column and keeps the sand moist and cohesive. (**B, C**) A typical nest at the start and termination of an experiment.

## Results

### Nest design and experimental setup

All experiments were conducted within two-dimensional nests (*Figure 1*, Materials and methods), nest sizes are reported as areas, but these can be readily translated into volumes by multiplying by the thickness of the setup (0.8 cm). We performed two kinds of excavation experiments: *colony-maturation* experiments that follow nest excavation alongside colony growth and *fixed-demographics* experiments that track digging by already mature colonies.

We initiated colony-maturation experiments by introducing a single mated queen and several brood items ($n = 5$, across all experiments) at random positions on the soil layer of the nest. This allowed us to capture nest excavation and colony development from their very first stages. In all instances where a collapse resulted in the queen's death or her being irreversibly trapped in the nest, the experiment was excluded from analysis starting from the point of the collapse, as such events did not reflect normal colony dynamics. Overall, colony-maturation experiments ($n = 22$) spanned up to 6 months (see *Appendix 1—table 1*; Experimental details, Materials and methods) during which the population size and nest area were recorded. The number of experiments decreased with time: for example, on day 130, we only had 13 ongoing excavation experiments ($n = 9$, collapses).

Fixed-demographics experiments ($n = 37$) included colonies with known age demographics. In preparation for these experiments, we tracked the age of individual workers over the course of 10 months (Materials and methods). We then extracted the queen alongside several workers of a given age and transferred them into the digging setup. Fixed-demographics experiments spanned six conditions which varied by worker age (young vs. old workers) and group size (5, 10, and 15 ants, including the queen). These experiments allowed us to track nest excavation as the ants were introduced into the setup and to further quantify the response to an induced catastrophic collapse.

### Nest and colony development from the founding queen

Colony-maturation experiments allowed us to track the concurrent growth of the colony and its nest. In all colonies, population size increased logistically and saturated at a value that was highly variable across colonies (typically between 5 and 20 ants; *Appendix 1—figure 1*; *Appendix 1—table 1*). As the population expanded, so did the nest (*Figure 2B*). The nest excavation was commenced by the founding queen, which dug a mean area of 23.8(±1.7) cm² ($n = 22$; unless noted otherwise, means are presented as mean ± standard error). Once the first workers emerged from the pupae, the queen ceased digging (personal observation). However, these first workers did not dig (see kink in *Figure 2B, C*) possibly because the initial area dug by the queen led to low ant densities (number of ants/area) at this stage. As more workers emerged, the area of the nest increased and varied considerably across

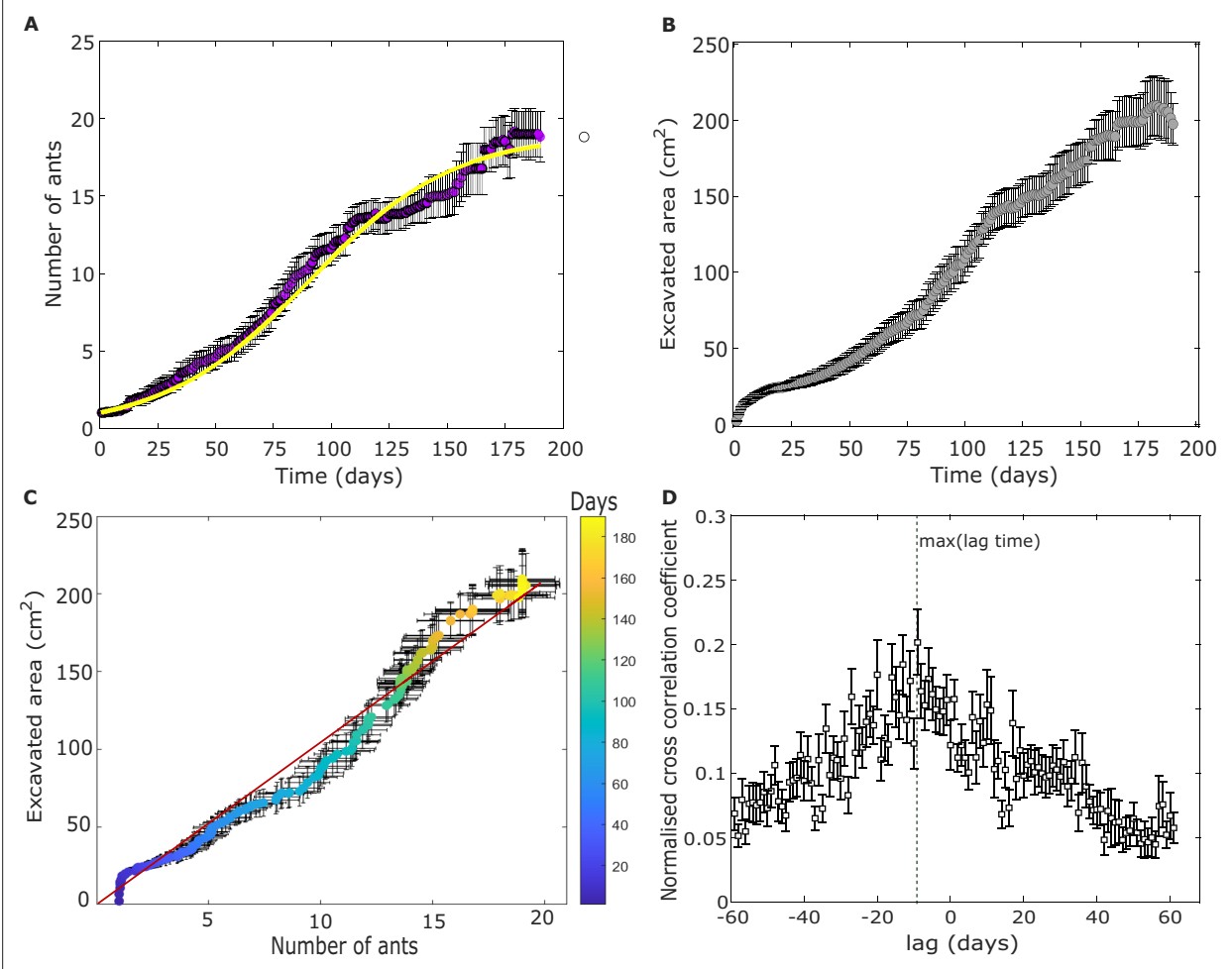

**Figure 2.** Excavation and population dynamics during colony maturation. Temporal change of the number of ants (**A**), and excavated area (**B**) starting from a single queen. Solid violet and gray circles show the mean number of ants and excavated area across the 22 colony-maturation experiment. Error bars depicted in black represent the standard error (here, and in all other figures). In (**A**), we fit a population growth model (yellow continuous line) using the number of ants as averaged over all colonies (Materials and methods). (**C**) The average excavated area is plotted against the corresponding population size. The solid red line depicts a linear fit $y = 10.71x$, $R^2 = 0.97$. The color bar represents the time in days. (**D**) The normalised cross-correlation coefficient between the digging rate and population growth rate is shown. The digging rates are correlated with earlier (negative lags) population change with maxima at lag '−10', rather than with the current (zero lag) or future (positive lags). Solid squares correspond to the mean cross-correlation coefficient. The vertical green line highlights the lag time with the maximum mean cross-correlation coefficient.

nests (*Appendix 1—figure 1*). For instance, halfway through the experiment duration, around day 95, the excavated area ranged from 38.41 to 173.28 cm$^2$ with a mean excavated area of 100.51(±11.54) cm$^2$ ($n$ = 15). At day 190, excavated areas ranged from 149.87 to 234.65 cm$^2$ with a mean excavated area of 197.47(±13.8) cm$^2$ ($n$ = 6). The number of active experiments decreased over time since experiments were terminated as specified above.

Previous works on already mature colonies revealed a linear relation between population and nest size (*Rasse and Deneubourg, 2001*; *Buhl et al., 2005*). We find that this relationship persists in cases where colonies and nests are allowed to co-expand (*Figure 2C*). A linear fit ($R^2$ = 0.97) indicates that the colonies regulate an area of approximately 11 cm$^2$ per each ant. We further quantified the relationship between the area per ant and experimental duration from the colony-maturation experiments using two alternative definitions of day 1: (a) the start of the experiment and (b) the emergence of the first worker. We found that the area stabilized at 11.1 (±1) for scenario (a) and 11.6 (±1.15) for scenario (b) (*Appendix 1—figure 2*). Deviations from this fit at early times result from the initial area excavated by the queen.

All the results stated so far deal with averages over different colonies. These do not allow us to address the causality of the regulation depicted in *Figure 2C*. To test whether colony expansion triggered changes in nest area or if it was the other way around, we tracked these concurrent processes one colony at a time. In all experiments, the initial population growth is followed, 15(±12) days later, by an increase in excavated area. Moreover, in some of the colonies, these dynamics repeat themselves, leading to a step-like pattern (*Appendix 1—figure 3*). This staggered pattern suggests that the nest is excavated in separate digging episodes, possibly instigated by a population increase. Cross-correlation analysis between the digging rate and the population growth rate (*Figure 2D*) indeed reveals that an increase in the population size is followed by an increase in nest area on a time scale of 1 week.

Our results suggest that population size controls nest area and not vice versa. As newly born ants join the colony, the size of the nest increases accordingly. It is, however, not clear which ants do the actual sensing or digging. One possible explanation is that, although all ants are capable of digging, it is primarily the newly emerged ants who perform this task. In this case, nest expansion would lag behind colony growth due to two delays: first, the time needed for young ants to mature enough to begin digging, and second, the physical time required to excavate additional space (e.g., around 10 days). This mechanism could eliminate the need for ants to assess overall colony density, as each new group of active workers simply enlarges the nest as they become ready. An alternative possibility is that all ants, regardless of age, respond to increased density by initiating excavation. In that scenario, nest expansion would follow more immediately after the emergence of new individuals, making delays less prominent (*Buhl et al., 2005*; *Buhl et al., 2004*; *Rasse and Deneubourg, 2001*).

## Fixed-demographics experiments reveal age-dependent nest excavation

In 'Colony-Maturation' experiments, ants emerge from their pupae inside the thin confines of the setup. This prevents us from marking them by age. To test how an ant's age affects her propensity to dig, we conducted shorter 'Fixed-Demographics' excavation experiments with fixed population sizes (5, 10, or 15 ants including a queen) where all workers are of similar age (either 'young' (40 ± 16 days) or 'old' (1712 ± 0)) (*Appendix 1—figure 4*). These ages represent functionally different life stages—the younger group had completed about 25% of their expected lifespan at the start of the experiment, while the older group had lived through most of theirs (*Richardson et al., 2021*; *Vonshak and Shlagman, 2009*). This fourfold age difference allowed us to compare excavation behaviors across fundamentally different phases of adult life. These experiments (Materials and methods, *Appendix 1—table 2*) further allowed us to study excavation under conditions of extreme area shortage. Under our

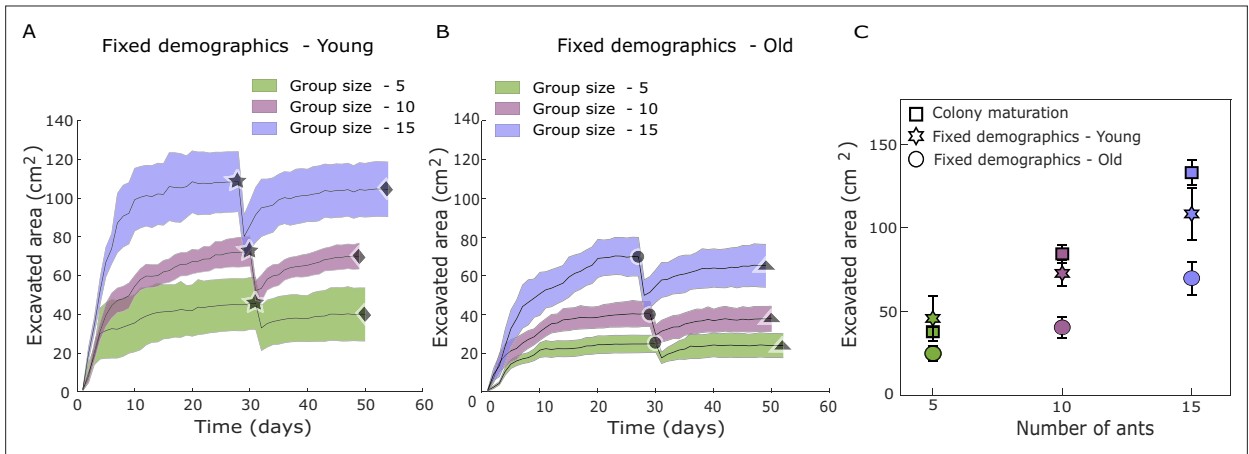

**Figure 3.** Excavation dynamics for similar-age groups. (**A, B**) The area excavated by the fixed-demographics young and old colonies with a colony size of 5, 10, and 15 ants. Solid stars and circles indicate the mean excavated area on the day of collapse. Diamonds and triangles indicate the mean area recovered after the manual collapse. (**C**) The mean area excavated by colonies of different group sizes from the colony-maturation and fixed-demographics experiments is shown. Squares (colony maturation), stars (fixed-demographics young), and circles (fixed-demographics old) correspond to the mean area excavated by colonies of a particular group size. Shaded error bars and error bars correspond to the SEM.

protocol, such conditions occur twice: as the ants are initially introduced to the setup and then again after we induce a collapse.

Following the introduction of the ants into the setup, the nest area grew logistically and stabilized after about 3 weeks independently of age or group size (*Figure 3A, B*). We found a significant difference in the maximum area excavated by same-sized cohorts as a function of worker age, with the younger ants excavating larger areas when compared to the older ants (*Figure 3A, B*—unpaired Wilcoxon signed-rank test significant for group size: 10; p = 0.014, 15; p = 0.042). Once the fixed-demographics nests reached a steady-state area, we artificially collapsed 25–30% of the steady-state area (star—*Figure 3A* and circle—*Figure 3B*, Materials and methods—Nest artificial collapse). We found that irrespective of age, ants dug after the collapse. Moreover, colonies ceased digging when they recovered (93 ± 3)% of the area lost by the manual collapse (diamonds and triangles—*Figure 3A, B*).

We compared the saturated excavation areas (pre-collapse) from fixed-demographics experiments (young and old groups) with those from colony-maturation experiments of the same colony sizes (*Figure 3C*). We find that, for a given age cohort (young or old), the saturation areas increase linearly with colony size (GLMM, $F_{(35,37)}$; p < 0.0001) (*Figure 3C*; *Appendix 1—figure 7*). The observed proportional scaling between excavated area and group size aligns with previous studies, even though those studies did not explicitly account for age demographics (*Buhl et al., 2005*; *Rasse and Deneubourg, 2001*; *Buhl et al., 2004*). After normalizing the pre-collapse excavated area by group size for both young and old colonies, we found no significant difference in area per ant across group sizes (*Figure 3*, *Appendix 1—figure 5*). This indicates that the excavated area per ant remains relatively constant within each demographic group. Furthermore, the area excavated by the young cohorts was comparable to that excavated by naturally maturing colonies when they reached the same population size (Tukey's HSD; group size: 5, p = 0.61; group size: 10, p = 0.46; group size: 15, p = 0.20). This holds even after excluding the area excavated by the queen from the colony-maturation experiments (*Appendix 1—figure 5B*). Digging rates (change in excavated area per day) for all group sizes in fixed-demographics experiments (young and old ants) peak within the first 7 days, then asymptotically decay to near-zero levels as nest excavation approaches saturation (*Appendix 1—figure 8*).

Normalizing saturation areas and digging rates by the number of ants allowed us to obtain the single ant digging rate as well as the target area per ant (this is the reciprocal of ant density which, for short, we refer to as the target area from here on). We find that while the target area of an ant depends on her age (*Figure 4A*), the post-collapse maximum digging rates revert to the pre-collapse levels, revealing an age-independent maximum digging rate (*Appendix 1—figure 6A,B*), over the first 200 days of an ant's life (*Appendix 1—figure 7B*).

## Age-dependent model predicts minimal digging by older ants

In the previous sections, we showed that in fixed-demographics experiments, younger ants excavated a significantly larger nest area compared to older ants (*Figure 3C*). This difference emerged despite similar temporal patterns in digging rates across age groups, with excavation activity peaking within the first 7 days before asymptotically decaying as nest expansion approached saturation (*Appendix 1—figure 8*). We make a key important assumption in our model: ants rely on local cues during nest excavation, and individuals cannot distinguish between the fixed-demographics and colony-maturation conditions. This implies that the age-dependent target area identified in the fixed-demographics experiments should also account for the excavation dynamics seen in the colony-maturation experiments. This assumption facilitates a re-analysis of the colony-maturation experiment, allowing us to estimate which ants are involved in nest excavation during normal nest growth and investigate its implications.

To do this, we present a modified version of the constant density regulation model of nest excavation (*Buhl et al., 2005*; *Buhl et al., 2004*). In the constant density regulation model, all ants dig toward the same target area, regardless of their age. In contrast, based on our findings from the fixed-demographics experiment, our modified model incorporated a dynamic component; the digging rate of each ant now depends on how similar the current area per ant ($a(t)$, which is one over the density) is to her age-dependent target (*Equation 1*).

$$\frac{da}{dt} = r\left(1 - \frac{a}{a_{(age)}}\right) \qquad (1)$$

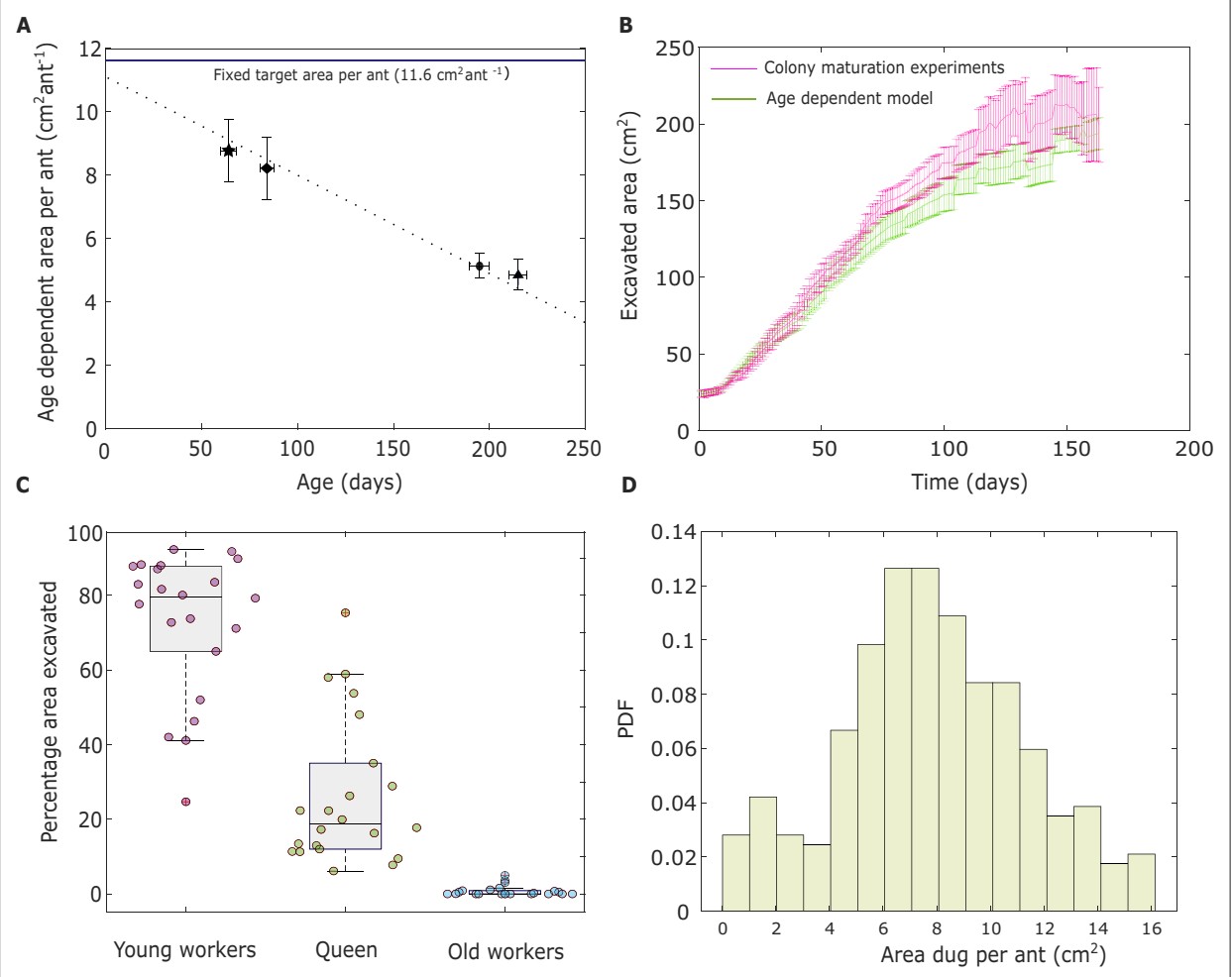

**Figure 4.** Age-dependent nest excavation. (**A**) The target area per ant is represented as a function of the age of individual ants (dotted line). Target areas decrease linearly with the ant age ($y = -0.032x + 11.22$, 95 % CI (Slope: (−0.035,−0.027), Intercept: (10.53,11.91)), $R^2 = 0.96$). The fixed target area ($11.6 \text{cm}^2\text{ant}^{-1}$) calculated from the colony-maturation experiment is shown with a solid blue line. (**B**) The mean excavated area as predicted by the age-dependent digging model is shown alongside the excavated area from the colony-maturation experiments. Error bars in (**A, B**) represent the SEM. (**C**) Total excavated area during normal colony growth as broken down by young ants, old ants, and the queen as predicted by the age-dependent model. The age threshold for young ants was set to be 56 days. Solid circles represent the area excavated by ants across 22 experiments. Boxes and median lines represent inter-quartile range and median values, and whiskers represent minimum and maximum values of data within 1.5-fold of the inter-quartile range. (**D**) The area excavated by each worker ant during normal colony growth as estimated by the age-dependent model.

where $a_{age}$ is the target area of a specific age (**Figure 4A**) and the basal digging rate constant is $r = 2.2 \pm 0.43 \frac{\text{cm}^2}{\text{ant} \cdot \text{day}}$ (as above). The instances where the current area per ant ('$a$') is higher than the steady-state area per ant $a_{(age)}$ resulted in a negative digging rate. While there were very few instances where ants filled up the excavated regions, they were extremely rare. Therefore, in the model, negative digging rates were taken to be zero. In the model, we assume that the ants rely on local cues to perform nest excavation, and they are unaware of the experimental condition (e.g., fixed demographics or colony maturation).

The total excavation rate $\frac{dA}{dt}$ for a colony that includes $'N'$ ants of different ages can be obtained by summing over the area contributions ($a$) from each ant:

$$\frac{dA}{dt} = \sum_{i=1}^{N} r \left( 1 - \frac{a}{a_{(age)}} \right) \qquad (2)$$

Since we recorded the population sizes in all colony-maturation experiments, and mortality events were rare, we could readily deduce the entire age demographics of the colony for each experiment

(SI section: Age quantification from colony-maturation experiments, *Appendix 1—table 3*). Using this data, we could integrate *Equation 2* over each experiment to obtain the total excavated area.

The models' results and their comparison to colony-maturation experiments are shown in *Figure 4B*. In both model and colony-maturation experiments, 'day 1' corresponds to the day on which the first workers emerge. The area on this day corresponds to the area excavated by the queen. We find that the age-dependent model fits the excavation dynamics observed in the colony-maturation experiments. Furthermore, we simulated an age-independent density model (*Appendix 1—figure 9*) using a constant target area per ant ($11.6 cm^2 ant^{-1}$, *Appendix 1—figure 2*), as obtained from the colony-maturation experiments, rather than an age-dependent area target that decreases with age (*Figure 4A*). While the age-independent model provides a reasonable fit for the colony-maturation experiments, it fails to align with the results of the fixed-demographics experiments. This discrepancy arises because the fixed-demographics experiments with young and old groups revealed a significant difference in the excavated area, despite having the same group size. This finding demonstrates that the area per ant is not a fixed value, as assumed in the age-independent density model.

Notably, the target area fitted for the age-independent model closely approximates the empirically measured age-dependent target when extrapolated to very young ants (*Figure 4A*). This raises the hypothesis that, in the colony-maturation experiments, the youngest ants are responsible for the majority of the digging. Thus, in light of our assumption that ants rely on local cues and do not distinguish between experimental contexts, these findings direct us toward an age-dependent model to explain the excavation dynamics in the colony-maturation experiments.

Since the target area is age-dependent and ants in the colony vary in age, not all ants perform the same amount of digging. By integrating *Equation 2* from the age-dependent model, we estimated the contribution of each age group to the total excavated area. We set an age threshold at 56 days (mean + standard deviation of the age of the younger cohort) in the age-dependent model to categorize the excavated area contribution between the young and old ants. This allowed us to quantify the area dug by the young ants, old ants, and the queen in terms of their contribution to the overall excavated area. We hypothesize that, as young ants have a lower digging threshold, they would perform the majority of the digging. Indeed, our model suggests that after the initial excavation by the queen, younger ants performed the vast majority of the digging (*Figure 4C*). The area excavated by older ants (age of 57 days and above) was nearly negligible. Thus, the age-dependent model provides indirect evidence indicating that younger members of the colony predominantly carry out nest excavation.

Furthermore, using the age-dependent model, we quantified the area dug by each ant for the colony-maturation experiment (*Figure 4D*), and analyzed the percentage contribution of each histogram bin to the excavated area (provided in *Appendix 1—table 4*). We found that 73.76% of the total excavated area ranged between 5 and 13 cm². This indicates that most ants dug within this range rather than exhibiting extreme variations. Additionally, the mean excavation amount is 7.84 cm², with a standard deviation of 3.44 cm², meaning that most values fall between 4.4 and 11.28 cm², which aligns well with the 5–13 cm² range. Since the majority of the excavation is concentrated within this narrow interval, and the mean is well centered within it, this suggests that ants excavated more or less the same amount, rather than forming distinct groups with highly different excavation behaviors.

These findings suggest that while all ants can dig and will do so following catastrophic events (see *Figure 3*), the regular nest expansion is primarily carried out by young ants.

## Structural classification of nests

The colony-maturation and fixed-demographics experimental conditions differed not only in the total volume of the nest but also in its architecture. We compared nest architectures by segmenting raw nest images into chambers and tunnels (see SI section: Nest skeletonization, segmentation, and orientation). Chambers were identified as flat, horizontal structures, while tunnels were narrower and more vertical in orientation (*Figure 5*, *Appendix 1—figure 10*, SI section: Nest skeletonization, segmentation, and orientation) (*Tschinkel, 2005*; *Buhl et al., 2004*; *Miller et al., 2022*; *Banavar et al., 1999*).

We found that the colony-maturation colonies exhibited a 'top-heavy' design (*Tschinkel, 2005*; *Tschinkel, 2013*; *Tschinkel, 2015*): large chambers (marked in yellow in *Figure 5A*) comprise most of the nest area at the top part of the nest (*Tschinkel, 2015*). Tunnels generally branch out from the large chambers and mostly lead to the bottom parts of the nest. The wide tunnels are primarily seen

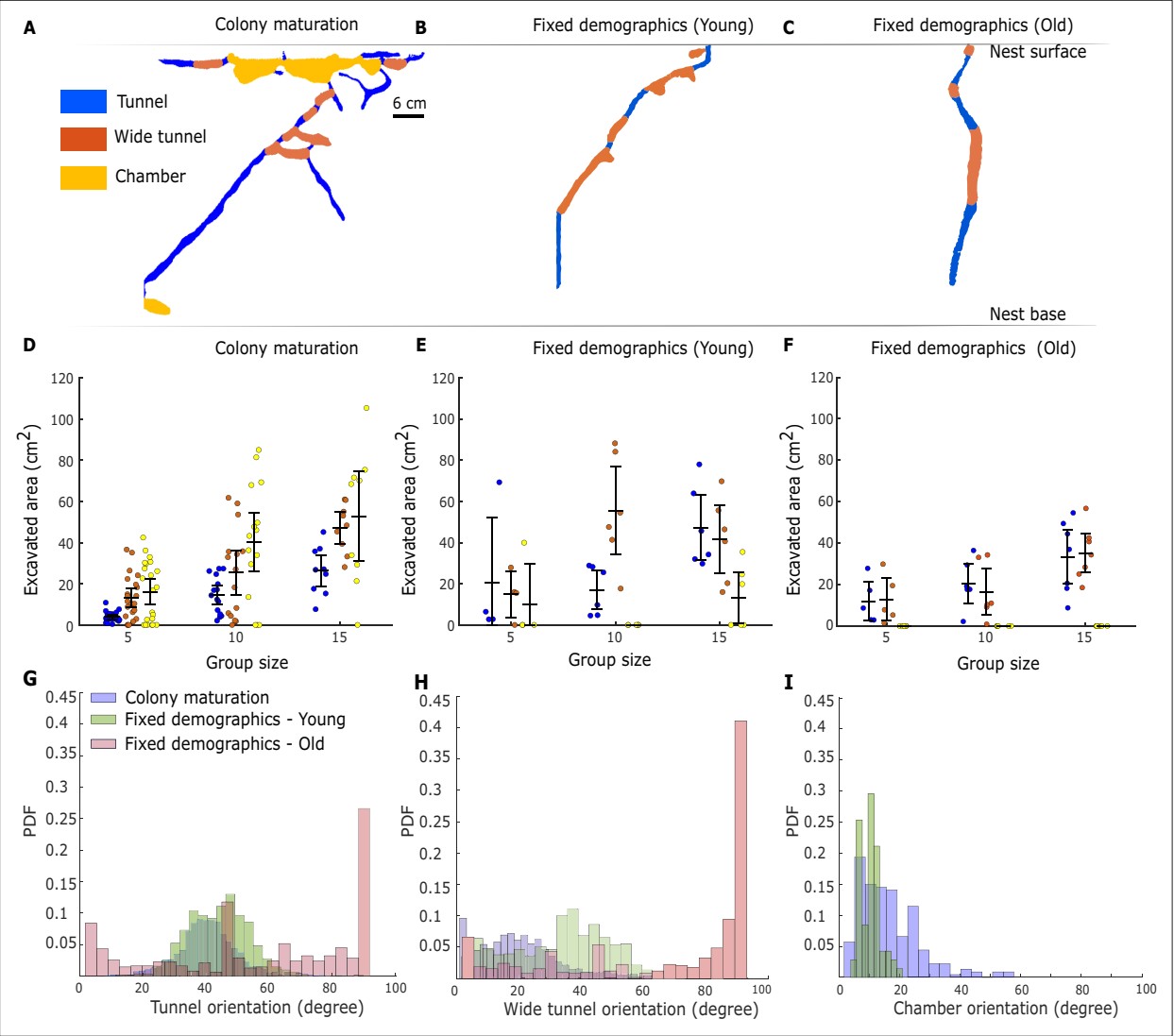

**Figure 5.** Structural classification of nests. Panels represent the nests excavated from the colony maturation (**A**), fixed-demographics young (**B**), and old (**C**) nests in terms of the tunnel, wide tunnel, and chambers for a colony size 15. (**D–F**) The contribution of tunnels and chambers to the total excavated area for a fixed group size across all the experimental conditions. Solid circles represent individual data points, and error bars represent 95% CI of the mean. (**G–I**) Angular orientation of tunnels and chambers across experimental treatments.

originating from the tunnels and are oriented parallel to the chambers (*Figure 5A*). In the fixed-demographics experiments, the nest structure mostly lacked large chambers, unlike the colony-maturation experiments (*Figure 5B–F*). This could be attributed to the absence of a young queen prone to dig and/or to the absence of brood.

We analyzed the orientation of all tunnels and chambers relative to the direction of gravity. We found that in the colony-maturation and fixed-demographics young experiments, the bulk of tunnels and wide tunnels are oriented between (30–60°). Surprisingly, in the fixed-demographics old colonies, the orientation of tunnels ranged from (70–90°) (*Figure 5*, *Appendix 1—figure 11*). Thus, the tunnels and wide tunnels in the colony-maturation experiments are similar to the fixed demographics of young colonies, but not the old ones.

Maturing colonies are composed of ants of different ages. Nevertheless, colony-maturation experiments displayed tunnel orientation similar to those in the nest of young fixed-demographics cohorts. This supports our prediction of the age-dependent model that younger ants primarily undertake digging tasks within maturing colonies.

## Discussion

Group size is a key determinant in the behavior of ant colonies. Groups of different sizes differ in their foraging strategies (*Beckers et al., 1989*; *Traniello, 1989*; *Detrain and Deneubourg, 2009*; *Narendra et al., 2008*; *Davidson, 1977*), division of labor (*Ulrich et al., 2018*; *Mersch et al., 2013*; *Hölldobler and Wilson, 1990*; *Ravary et al., 2007*; *Jeanson and Fewell, 2008*; *Robinson, 1992*; *Beshers and Fewell, 2001*; *Jeanson et al., 2007*), and communication networks (*Pacala et al., 1996*; *Pinter-Wollman et al., 2014*; *Pinter-Wollman et al., 2011*; *Fewell, 2003*). It is less clear what happens in the expansion period that follows colony foundation, when group size is constantly on the rise. A growing colony may exhibit collective patterns that differ from a fixed group of the same size. This is especially relevant if the system preserves long-term memories such as the stigmergic alteration of the environment evident in nest construction (*Khuong et al., 2016*; *Buhl et al., 2004*; *Deneubourg and Franks, 1995*; *Theraulaz and Bonabeau, 1995*; *Theraulaz and Bonabeau, 1999*; *Toffin et al., 2010*).

To date, the overwhelming majority of studies on nest excavation in ants were short-term experiments performed on a mature group of fixed size (*Aguilar et al., 2018*; *Buhl et al., 2005*; *Buhl et al., 2004*; *Toffin et al., 2009*; *Toffin et al., 2010*; *Rasse and Deneubourg, 2001*; *Franks and Deneubourg, 1997*; *Tschinkel, 2015*; *Kwapich et al., 2018*). As we have shown, these short-term studies likely address construction in emergency scenarios where there is a huge deficit in available space. Studying animal behavior in ethologically relevant conditions is most valuable for being able to correctly capture the nature of these animals and is crucial for deciphering the underlying meaning of their behavior. Furthermore, testing animals in the wrong context/conditions could lead to misleading conclusions (*Fonio et al., 2009*; *Fonio et al., 2012b*; *Olsson et al., 2003*; *Merali et al., 2003*; *MacLennan, 1990*; *Fonio et al., 2012a*).

The current study follows ant colonies over extended periods as they mature from a single reproductive individual. We find significant quantitative and qualitative differences between nests constructed during the colony-maturation experiments, which mimic natural colony development from a single queen, and nests constructed in the context of an emergency. The differences include the nest volume, structural layout, tunnel orientation, and the demographic identity of ants involved in excavation. Colonies from the colony-maturation experiments excavated the largest nest volumes for a given group size, followed by fixed-demographics young and then old colonies. Chambers were consistently present in colony-maturation nests, sparse in young, and nearly absent in old colonies. The orientation of both tunnels and wide tunnels was similar in the colony-maturation and fixed-demographics young experiments, but it was different in the old colonies.

A main advantage of social insects is their ability to allocate different tasks among individuals. A well-known example includes age-polyethism wherein ants perform age-specific tasks (*Hölldobler and Wilson, 1990*; *Ravary et al., 2007*; *Jeanson and Fewell, 2008*; *Robinson, 1992*; *Beshers and Fewell, 2001*; *Jeanson et al., 2007*; *Ulrich et al., 2018*; *Mersch et al., 2013*). Response thresholds that vary with age have been suggested as a mechanism for age polyethism (*Theraulaz et al., 1998*; *Bonabeau et al., 1996*; *Gove et al., 2009*; *Beshers et al., 1999*; *Richardson et al., 2011*; *Tarapore et al., 2010*). Previous studies of nest excavation indeed rely on a threshold mechanism that prompts ants to dig extra space when the nest becomes too crowded (*Buhl et al., 2005*; *Buhl et al., 2004*; *Rasse and Deneubourg, 2001*; *Aguilar et al., 2018*). Our results reveal that this target area of the group decreases linearly with age, such that young ants are more sensitive to space shortages. Thus, under normal growth conditions, nest volume remains proportional to the number of ants, ensuring a relatively constant area per ant. As a result, density thresholds for older ants are not exceeded, and excavation is predominantly carried out by younger ants. These findings have significant implications for the dynamics of nest volume.

In our colony-maturation experiments, we found that area per ant was highest when the workers were youngest, with values around 11.1–11.6 (±1–1.15). This aligns with observations from naturally growing nests, where newly eclosed ants dominate the population and nest volumes are relatively large. Supporting this, fixed-demographics experiments showed that the area excavated per ant declines linearly with worker age, indicating that the youngest ants contribute most to excavation. Notably, the target area we fit for the age-independent model (11.6 ± 1.15) closely matches the extrapolated value for very young workers (*Figure 4A*), reinforcing the idea that young ants are the primary excavators during early colony growth. In contrast, during events like collapses or displacement, when space is urgently needed, ants of all ages participate in excavation.

We highlight that *C. fellah* queens can live up to 20 years, while worker lifespan is significantly shorter, though exact data for this species are lacking (**Vonshak and Shlagman, 2009**; **Richardson et al., 2021**; **Mersch et al., 2013**). However, studies on related *Camponotus* species (**Rossi and Feldhaar, 2021**) suggest that workers can live several months (≈1 year for workers, and ≈2 years for majors; personal observation). Since our observed worker lifespan (≈200 days) falls within this range, it likely represents a substantial portion of their natural lifespan, justifying their classification as 'young' and 'old'.

In our study, we adopted the simpler version of previously published age-independent excavation models, where individuals respond to local stigmergic cues such as encounter rates or pheromone concentrations, which serve as a proxy for the global nest volume (**Buhl et al., 2005**; **Buhl et al., 2004**). We minimally modified this model to include age-dependent density targets. According to our age-dependent digging model, each ant compares this perceived local density to its own innate age-specific digging threshold as quantified from the fixed-demographics experiments. If the perceived local density exceeds its age-dependent area threshold (indicating insufficient area), it digs; otherwise, there is no digging. This mechanism eliminates the need for cognitively demanding global assessment of the total nest volume or the overall colony population, a requirement for the saturation model (**Rasse and Deneubourg, 2001**).

Our age-dependent model demonstrates that the digging behavior in *C. fellah* is governed by a basal digging rate constant ($r$) modulated by the age-dependent feedback ($1 - \frac{a}{a_{(age)}}$). Crucially, we show that after a collapse, the maximum digging rates return to their pre-collapse levels, suggesting that this basal rate '$r$' represents an age-independent ceiling on how fast ants can dig, regardless of age or context (**Appendix 1—figure 6A, B**). Previous studies have demonstrated both homogeneous and heterogeneous workload distribution, with varying digging rates among ants (**Buhl et al., 2005**; **Buhl et al., 2004**; **Aguilar et al., 2018**; **Rasse and Deneubourg, 2001**). Studies showing heterogeneous workload distribution relied on continuous individual tracking of ants to quantify digging rates (**Aguilar et al., 2018**). However, this approach was not feasible in our current design due to the experimental durations of both our colony-maturation and fixed-demographics experiments. Additionally, sample size requirements naturally limited our ability to conduct continuous individual tracking during nest construction in our study. Thus, based on empirical measurements from our fixed-demographics experiments and supported by the age-independent post-collapse digging rates, we adopted a constant basal digging rate for simulating our age-dependent model—an assumption aligned with both prior literature and the collective dynamics observed in our system (**Buhl et al., 2005**; **Buhl et al., 2004**).

Although we find that a constant density model provides a reasonable fit for the excavated area from the colony-maturation experiments, the results from the fixed-demographics experiments suggest otherwise. Despite having the same group size, young and old colonies exhibited a significant difference in the excavated area, which would not be expected if a constant density model were in effect. Thus, in light of our assumption that ants rely on local cues and do not distinguish between experimental contexts, these findings direct us toward an age-dependent model for excavation behavior.

Digging behaviors that vary across the ants' lifetime influence not only the nest volume, but also its structure. During colony founding, excavation is performed solely by the queen, who typically creates several large chambers near the nest surface. These chambers, consistent with the density-target threshold model, delay the emergence of the first workers, who later assume the digging role. This transition shifts nest architecture from the initial top-heavy design to a more balanced structure as the excavation progresses. In particular, tunnel orientation varies with the age of the diggers, revealing a novel form of age polyethism. While previous studies focused on age-based task allocation (**Seeley, 1982**; **Robinson, 1987**; **Tofilski, 2002**; **Biedermann and Taborsky, 2011**; **Bernadou et al., 2015**; **Tripet and Nonacs, 2004**; **Nakata, 1995**), our results demonstrate that the nature of the task and its execution itself can shift with age. This behavioral plasticity introduces a new regulatory mechanism, distinct from the case-based performance differences driven by polymorphic species (**Kwapich et al., 2018**; **Gravish et al., 2012**).

Our results suggest that the nest reflects the cumulative effects of past events and conditions experienced by the colony. For example, in line with our experimental findings, if the colony expanded during a period with many young ants, more slanted tunnels are expected, indicating accommodation

for a growing population. Conversely, nest construction initiated by mature ants or a catastrophe leading to nest area reduction would result in newly constructed tunnels aligning more with gravity, running straight down. Thus, the nest structure could be treated as a historical record of the colony.

To conclude, ant colony demographics play a pivotal role in nest excavation. They affect the identity of the ants that do the digging, the dynamics of nest growth, the volume of the nest per population size, and the chamber and tunnel architecture of the nest. We find that age-polyethism provides a major means by which the colony exerts control over its nest in different scenarios.

## Materials and methods
### Study species
We chose to work with *C. fellah*, a common species in Israel of the ground-dwelling carpenter ants (7.8–17.2 mm body length, *Vonshak and Ionescu-Hirsch, 2009*). These ants are omnivorous, monogynous, and singly mated, with colony sizes ranging from tens to thousands of workers (*Mersch et al., 2017*; *Richardson et al., 2021*). We collected newly mated queens on the Weizmann Institute of Science campus after nuptial flights in March 2018. For the colony-maturation experiments, we used only queens with pupae that had already nurtured larvae to increase survival rates. We used colonies of group sizes (5, 10, and 15 ants) for the fixed-demographics experiments. We housed the single queen and fixed colonies in plastered nest containers and environmental conditions (climate-controlled chamber under controlled humidity (65%), temperature (27°C), and a light–dark cycle of 12 hr) before the experiments. We supplied the colonies with a food mixture (tuna, honey, eggs, vitamin mix, and water), water, and a 20% sucrose solution.

### Nest structure
The artificial nest structure consists of a 2D Plexiglas frame 60 cm wide, 80 cm high, and 8 mm deep made from 5 mm thick cast acrylic sheets. Six support blocks (2 cm) held the front and back frames together. We equipped the nests with a 5-cm wide water column connected by a series of 0.4-mm holes in the bottom that allow water transfer but not sand grains (*Figure 1A*). The top part of the frame was coated with Fluon (Sorpol) to prevent the escape of ants. A removable plexiglass lid perforated with numerous small (1 mm) holes for ventilation covers the top of the nest.

### Nest preparation
Each nest was filled 70 cm high with homogenous fine-grained sand (AccuSand 30–40, Unimin Co, USA) with grain sizes of 425–600 μm. Before the experiments began, water infiltrated gradually from the water column to the sand, slowly saturating the entire sand column from the bottom. Apart from wetting the sand, this action rearranged the sand grains in even more compact packaging and lowered the chance of spaces pening up in the column. On top of the sand layer, we placed a 1- to 2-cm layer of soil on the sand column, thus significantly reducing evaporation and the subsequent collapses. The soil was collected from the Weizmann campus orchard and sieved through coarse (2 mm) and fine (0.5 mm) sieves (Haver & Boecker, Germany) and was sterilized. We manually made a 30-mm long and 8-mm wide hole in the middle of the solid layer (seeding point) to induce a digging behavior.

### Nest maintenance
We provided food to the ants ad libitum through three separate tubes containing water, 20% sucrose water, and protein food. The protein mixture included egg powder, tuna, prawns, honey, agar, and vitamins. Each of the three tubes was filled with 5 ml of their respective contents and sealed with a cotton stopper to prevent overflow. The tubes were positioned at a slight angle and connected using a custom-made plexiglass adapter to facilitate the flow of liquids. These tubes were replenished once depleted and regularly replaced once the nest maintenance was carried out bi-weekly.

### Experimental overview
The study included 22 colony-maturation and 37 fixed-demographics experiments (fixed-demographics young, group size 5, *n* = 4, group size 10, *n* = 6, group size 15, *n* = 6); fixed-demographics old (group size 5, *n* = 6, group size 10, *n* = 7, group size 15, *n* = 8) of ant nest excavation. We placed the founding queen on top of a 60-cm wide, 70-cm high, and 8-mm thick soil-sand column in each nest.

We photographed the nest development every 1–3 days. During each photography session, four pictures of the nest were taken, with a 2-s interval between each. The experiments were terminated once the queen died or became trapped following a collapse.

## Fixed-demographics colonies

Queens used for the fixed-demographics experiments were collected during the nuptial flight of May 2020. Forty-five newly mated queens were reared for the fixed-demographics experiments, and 37 colonies that attained the required colony sizes needed for the fixed demographics were chosen for the experiments. The colonies were maintained with a 12-hr light/dark cycle, room temperature of 25 ± 2°C, and 60% humidity. Each freshly born ant was marked using a unique three-color combination of acrylic paint on the head, thorax, and abdomen to determine the age. We checked each colony at least once in 5 days to check whether new workers emerged. The marked worker's age was recorded and updated in due course. In the fixed-demographics experiments, we separated colonies into two different demographics: Young and Old. The young and old colonies had workers with an average age of 40 ± 16 and 171.56 ± 20 days, respectively. Error represents the standard deviation of the mean.

## Experimental conditions

We maintained a stable environmental condition in the experimental room by maintaining a 12-hr light/dark cycle and room temperature of 25 ± 2°C. Daily variations were 2–3°, usually 24–27°C. More significant deviations ranging at most 20–30°C did occur sporadically due to technical problems, but they did not last more than a few hours at a time. Humidity was more challenging to maintain, and it fluctuated in the range of 40–70%, averaging around 60%.

## Nest photography—data collection

A photography sequence of different exposures recorded the nest development in a measurement. The nests were placed one meter away from the camera against a diffuser-covered LED backlight. Images were initially taken with Canon EOS 800D DSLR cameras with a 24 MP resolution and fitted with EF-S 18–55 mm f/4–5.6 IS STM lenses. We controlled the cameras remotely using a USB cable through the 'DigiCamControl' software (Istvan, 2019) with the photography sequence programmed using MATLAB CameraController (Serge, 2019). The photography sequence consisted of four photos taken at 2-s intervals.

## Nest artificial collapse

To induce manual collapses, we started by quantifying the nest area from the start of the experiment, as described in the Image processing and excavated area section to get the steady-state nest area. Once we made sure that the excavated area approached a steady state, we fixed that 25–30% of the excavated area should be collapsed. This calculation was based on image analysis that provided the area to be collapsed in square centimeters. To ensure accuracy, we used graph paper to trace the entire nest area, allowing us to map the overall structure to a scale that facilitated precise collapse induction. We then compared the nest area obtained from image processing with the graph paper tracings to verify consistency in both the steady-state area and the area designated for collapse. Next, we fixed the graph paper onto the Plexiglas of the nest, against a light background, enabling precise alignment of the nest skeleton with all excavated structures. The collapsed area was marked on the graph paper, ensuring precise targeting for collapses. The collapse was induced in the deepest excavated region of the nest, in the direction of gravity, using a small cylinder with a 2-mm diameter. We specifically chose to collapse the deepest region to prevent further structural degradation that could lead to unintended collapses. Additionally, this approach ensured that ants were not irreversibly trapped within collapsed sections, allowing them to continue excavating the nest without obstruction.

## Image processing and excavated area

The images were post-processed using the MATLAB Image Processing Toolbox. The first step was image registration, where we aligned nest images with the first image taken before introducing ants. The algorithm registers the images through a projective geometric transformation using the six support blocks in the frame as control points. After registration and cropping, we binarized the images to show only the excavated areas of the sand column. The first binarization step

involved subtracting the initial image from the processed image to reduce glare and noise. We then converted the RGB image into an intensity image by choosing the maximum value between the green or blue channels. Next, we used adaptive thresholding to produce BW (black and white) images of the nest architecture in which white pixels represented the excavated areas. Lastly, we made manual corrections to mark known excavated zones, which were not evident in the images, or remove other apparent noises. The excavated area refers to each Black and White image's white pixel area.

## Statistical analysis

Statistical analysis was conducted using the built-in functions in MATLAB 2021a. Tests were performed with a significance level of 0.05. For all statistical tests, we checked the necessary assumptions by plotting data histograms, distribution of model residuals, and checking for unequal variances.

## Acknowledgements

We thank Dr. Yuri Burnishev, Guy Han, Ezra Eliyahu, the physics workshop, and the physics electronics team for the technical support in making the nest frame and camera system. We also thank Dr. Aviram Gelblum, Dr. Lior Baltiansky, Dr. Atanu Chatterjee, Tabea Heckanthler, Dr. Yael Heyman, Michal Roitman, Yohai Nurenburg, Krishna Sindhu, and Nina Weinberger for the instructive comments during the project and the manuscript. OF thanks the Israeli Science Foundation (ISF grant 1727/20) for their support.

## Additional information

### Funding

| Funder | Grant reference number | Author |
|---|---|---|
| Israel Science Foundation | 1727/20 | Ofer Feinerman |

The funders had no role in study design, data collection, and interpretation, or the decision to submit the work for publication.

### Author contributions

Harikrishnan Rajendran, Conceptualization, Data curation, Software, Formal analysis, Supervision, Validation, Investigation, Visualization, Methodology, Writing – original draft, Project administration, Writing – review and editing; Roi Weinberger, Conceptualization, Data curation, Software, Formal analysis, Validation, Investigation, Visualization, Methodology, Project administration; Ehud Fonio, Conceptualization, Data curation, Formal analysis, Investigation, Methodology; Ofer Feinerman, Conceptualization, Resources, Supervision, Funding acquisition, Validation, Investigation, Methodology, Writing – original draft, Project administration, Writing – review and editing

### Author ORCIDs

Harikrishnan Rajendran  https://orcid.org/0000-0003-4795-1666
Ofer Feinerman  https://orcid.org/0000-0003-4145-0238

Reviewer #1 (Public review): https://doi.org/10.7554/eLife.100706.4.sa1
Reviewer #2 (Public review): https://doi.org/10.7554/eLife.100706.4.sa2
Author response https://doi.org/10.7554/eLife.100706.4.sa3

## Additional files

### Supplementary files

MDAR checklist

## Data availability

The data and code associated with this study are available at https://doi.org/10.6084/m9.figshare.31457431.

The following dataset was generated:

| Author(s) | Year | Dataset title | Dataset URL | Database and Identifier |
|---|---|---|---|---|
| Rajendran H, Weinberger R, Fonio E, Feinerman O | 2026 | Colony demographics shape nest construction in ants | https://doi.org/10.6084/m9.figshare.31457431 | figshare, 10.6084/m9.figshare.31457431 |

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

# Appendix 1

## Individual experiments—colony maturation

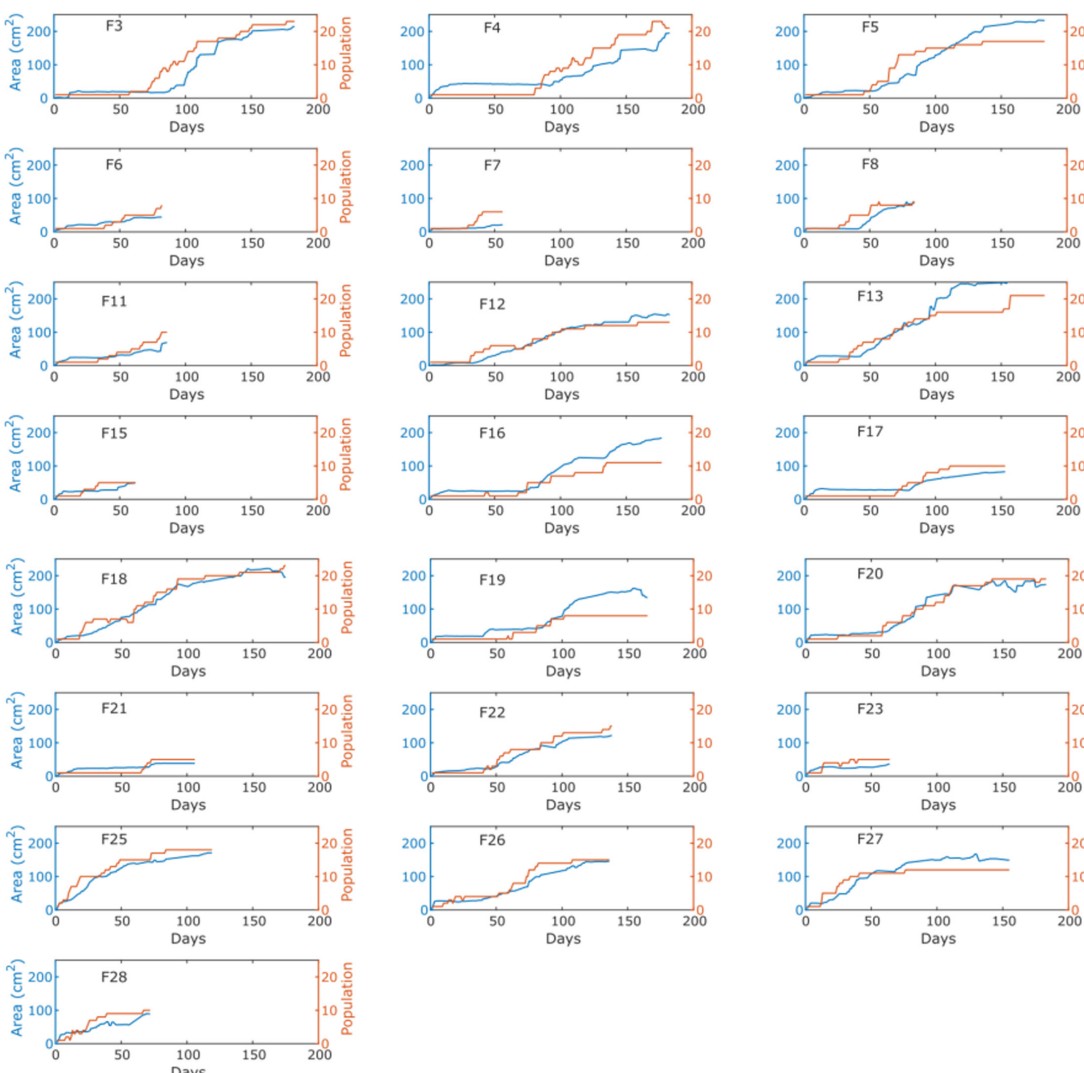

**Appendix 1—figure 1.** Individual experiments—area and population. The change of area and population across individual colony-maturation experiments (*n* = 22) is provided. The left *y*-axis shows the excavated area and the right *y*-axis shows the population.

## Area per ant from colony-maturation experiments

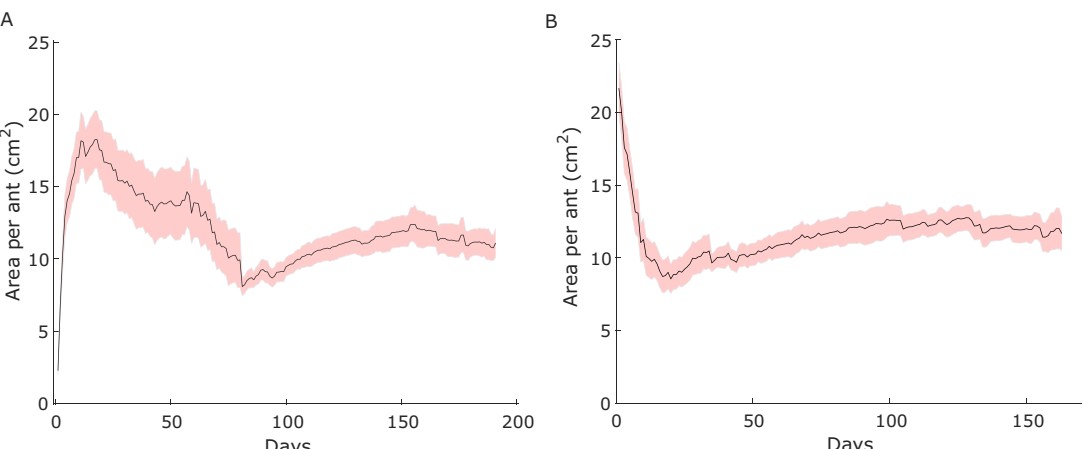

**Appendix 1—figure 2.** Temporal change of the area per ant. The temporal change of the area per ant from the colony-maturation experiments. The area per ant is calculated by dividing the excavated area by the number of ants in the experiment. In **A**, day '1' corresponds to the start of the experiment, whereas in **B**, day '1' corresponds to the day on which the first workers were born. The area per ant stabilizes at 11.1(±1) for **A** and 11.6(±1.15) for **B**. Shaded error bar in both figures shows the SEM.

# Individual colony-maturation experiments reveal a step-like pattern

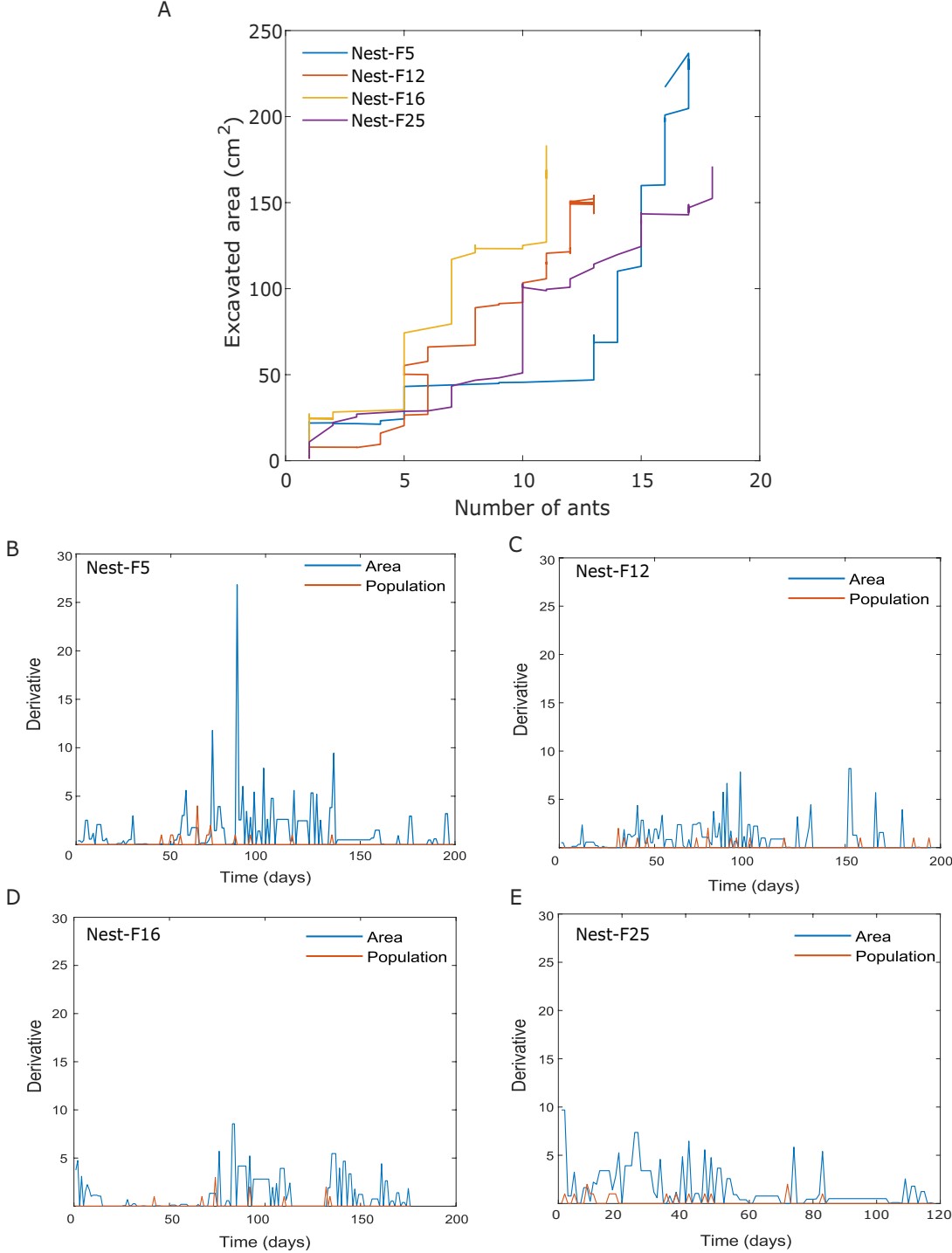

**Appendix 1—figure 3.** Step-like relationship between excavated area and population growth. (**A**) The excavated area is plotted against their corresponding population sizes across specific colony-maturation experiments where the step pattern was most evident. We can see a 'step' like the relationship between the variables. (**B–E**) It shows the change in area and population across the experimental duration in the four specific nests. Peaks correspond to the increase in area or population.

## Age distribution in fixed-demographics experiments

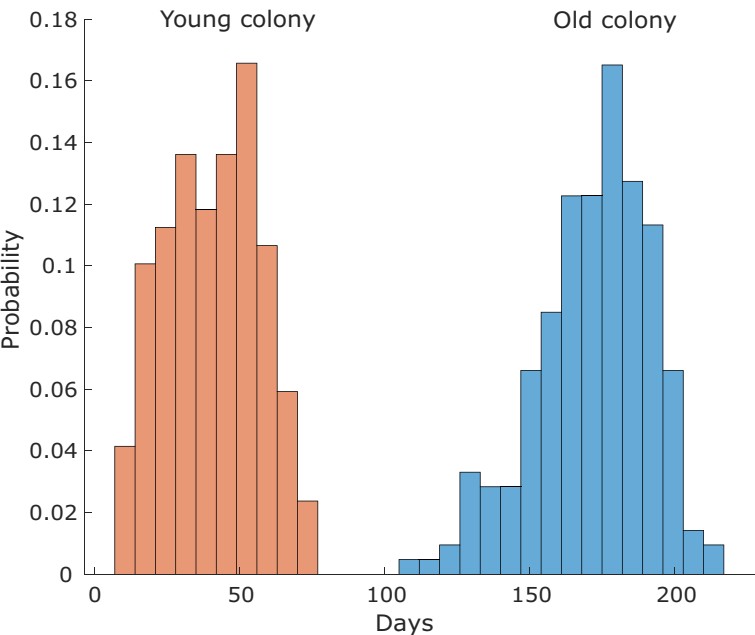

**Appendix 1—figure 4.** Age distribution from the young and old colonies. The age distribution of workers used for the fixed-demographics young and old experiments is shown as a histogram. The ants in the fixed-demographics young experiments had an average age of 40 ± 16 days, and old experiments had an average age of 171.56 ± 20 days. Errors represent the standard deviation.

## Excavated areas scale with group size, and young fixed-demographics colonies excavate similarly to colony-maturation experiments

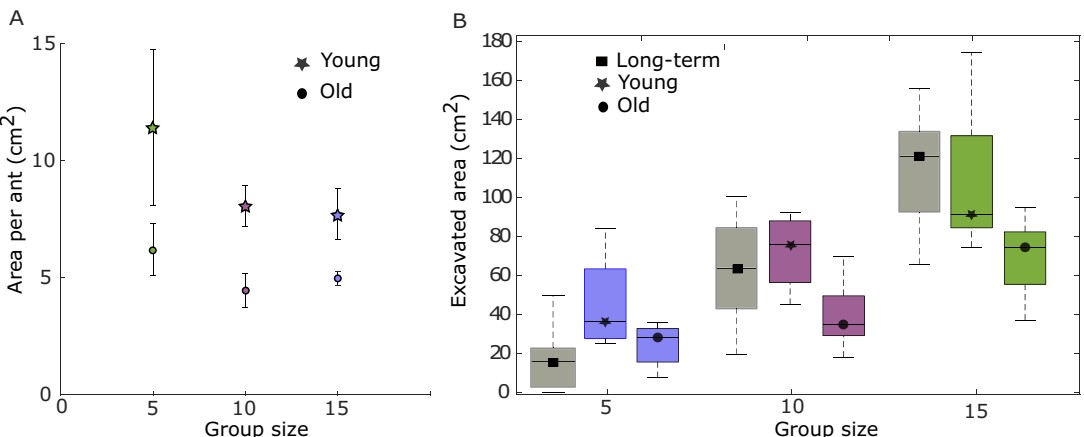

**Appendix 1—figure 5.** Excavated areas normalized by group size and the maximum excavated area in fixed-demographics experiments, excluding the area excavated by the queen. (**A**) The normalized excavated area per ant for both young (star) and old (circles) colonies shows no significant difference across group sizes, indicating that the total excavated area is proportional to group size. Error bars represent standard error. (**B**) The area excavated by the queen from the colony-maturation experiments was excluded and compared to the resultant area from the fixed-demographics young and old experiments. We find that the area excavated from the colony-maturation experiments is similar to the fixed-demographics young experiments.

## Maximum digging rate from fixed-demographics experiments

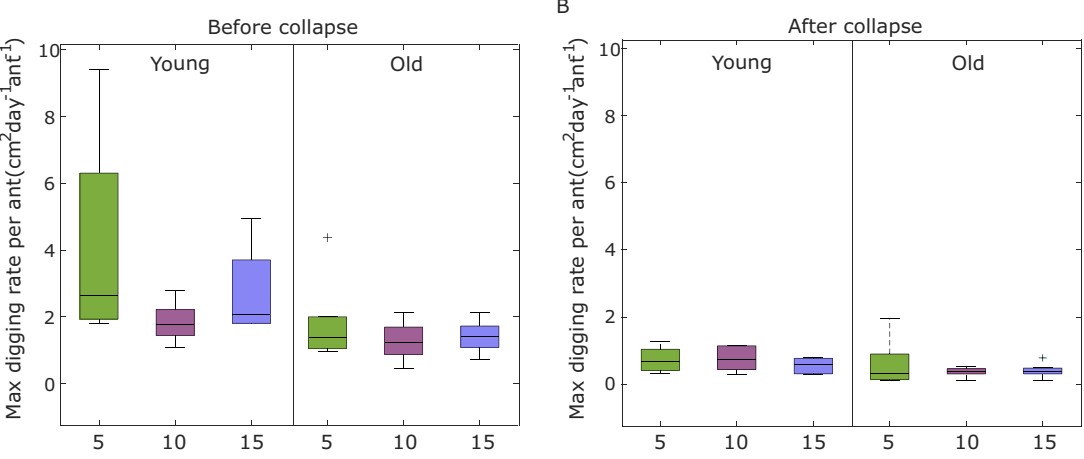

**Appendix 1—figure 6.** Maximal digging rate before and after collapse. The maximum digging per ant before (**A**) and after (**B**) the manual collapse events from the fixed-demographics experiments. The maximum digging rate per ant is calculated by normalizing the maximum digging rate (change in area per day) by the number of ants. We find that there is no significant difference before and after collapse events (Kruskal–Wallis test, $\chi^2(5,68) = 6.01$, p = 0.30).

## Excavated area and digging rate from fixed-demographics experiments

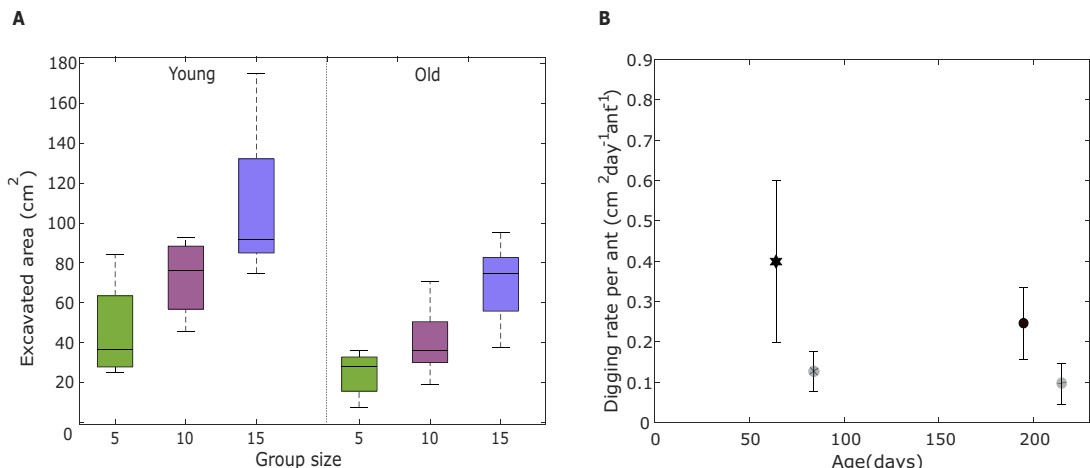

**Appendix 1—figure 7.** Excavated area and digging rate. (**A**) The excavated area across group sizes from the fixed-demographics young and old experiments. We performed a linear regression on the excavated area across different group sizes for the fixed-demographics young ($y = 6.35x + 0.73, R^2 = 0.43$) and old ($y = 4.57x - 0.73, R^2 = 0.57$). The positive slope indicates that the excavated area increases with group size for both the young and old colonies. (**B**) The digging rate per ant for different age categories pre- and post-collapse events is shown. We find that the digging rate per ant is age-independent. Error bars represent the SEM.

## Digging rates from fixed-demographics experiments before inducing manual collapses

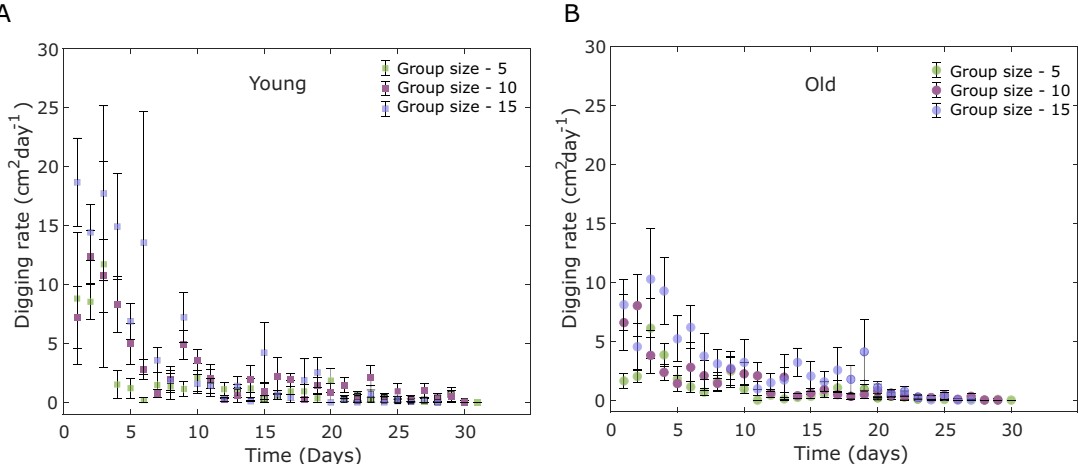

**Appendix 1—figure 8.** Digging rate dynamics. Digging rate from the fixed-demographics young (**A**) and old (**B**) experiments across three different group sizes. The digging rate is calculated as the change in the area per day. Error bars represent the standard deviation.

## Age-independent density model

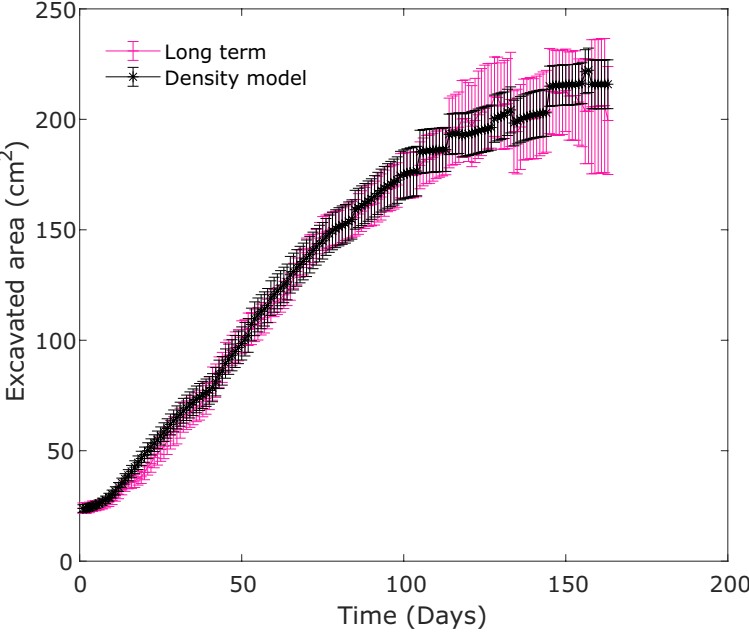

**Appendix 1—figure 9.** Density model. The progression of the excavated area based on the constant density regulation model and its comparison to the colony-maturation experiments. The constant density was fixed to be at $11.6 cm^2 ant^{-1}$. Error bars represent the SEM.

## Nest skeletonization, segmentation, and orientation

To estimate the total length of the nests, we created skeleton images of the binary images (*Appendix 1—figure 10A*). Initially, images were preprocessed with an averaging filter to smooth the shape edges. The skeleton was derived by MATLAB's, 'bwmorph' function using the 'thinning' morphological operator set to infinity. We further performed a 'Medial Axis Transform' (MAT) on the

nest skeleton, where every skeleton pixel is assigned a distance value to the closest boundary of the excavated area. Many natural nests can be clearly divided into chambers and tunnels, while in other species the division is unclear (*Tschinkel, 2015*). Moreover, previous studies define chambers as flat horizontal structures compared to vertical narrow tunnels. With no available data on the architecture of natural nests of *Camponotus fellah* ants to compare with, we segmented the nest into tunnels, wide tunnels, and chambers using the decision algorithm as described in *Appendix 1— figure 10*. On the inverse of the binary image, we assigned each pixel in the excavated area to its shortest distance to the edge. Those assigned with a distance of more than 10 pixels ($\approx 0.65\text{cm}$) were designated as seeds for large chambers. Next, a geodesic distance transform assigned pixels the distances to these seeds and outlined the chambers. This created separate blobs, groups of connected pixels in a binary image, for the two classes with a width threshold of 1.30 cm between tunnels and potential chambers. From the width distribution, we choose 3.15 and 1.6 cm as high and low thresholds. Blobs above the high threshold were assigned as chambers. Blobs under the low threshold and relatively flat (absolute value of orientation under 30°) and overall regular in shape are added to the first class of tunnels. The residual blobs were defined as wide tunnels. The shape regularity was estimated by blobs with less than 30% deviation from the mean of the MAT pixels. The resultant segmented images from the colony maturation, fixed-demographics young, and old colonies are provided in (*Figure 5A–C*). The orientation analysis was performed to quantify the angular orientation of all tunnels, wide tunnels, and chambers from the initial seeding point concerning the direction of gravity.

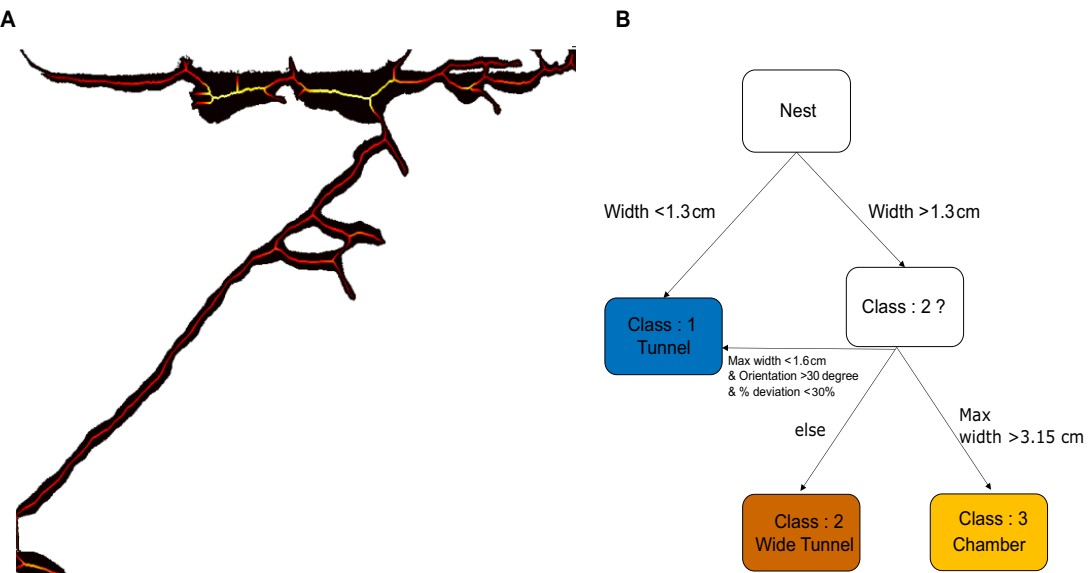

**Appendix 1—figure 10.** Nest segmentation algorithm. (**A**) Nest skeletonization—The medial-axis transform (MAT) image of the excavated area of a sample nest from the colony-maturation experiments. The warmer color of the skeleton represents a larger distance to the edge of the excavated structure, shown in black. (**B**) The decision tree used for the structural segmentation of the nest images into three classes: namely, (1) tunnels, (2) wide tunnels, and (3) chambers.

## Nest architecture

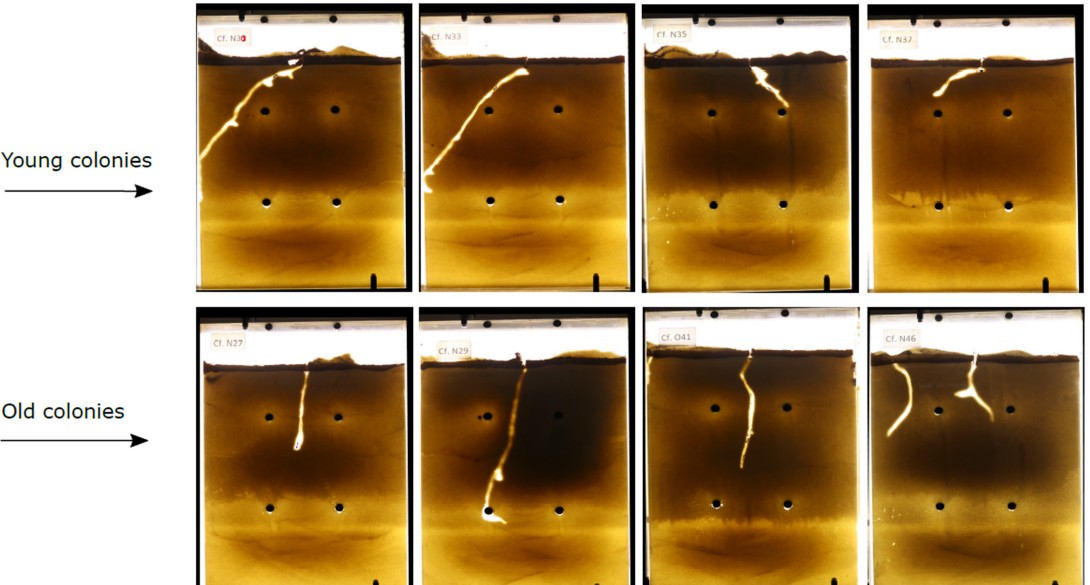

**Appendix 1—figure 11.** Nests excavated by colonies from the fixed-demographics treatments. The nest structures excavated by the young and old colonies from the fixed-demographics experiments are shown. The first row corresponds to the nest images from the young colonies, and the second row corresponds to the old.

## Colony-maturation and fixed-demographics experiments

**Appendix 1—table 1.** Details of colony-maturation experiments.

**Colony-maturation experiments**

| Nest-id | Start date | End date | Max. population | Max. area (cm²) |
|---|---|---|---|---|
| F3 | 23-03-2018 | 28-10-2018 | 22 | 214.44 |
| F4 | 30-03-2018 | 28-10-2018 | 22 | 291.29 |
| F5 | 08-04-2018 | 24-10-2018 | 16 | 233.39 |
| F6 | 05-04-2018 | 08-10-2018 | 7 | 44.35 |
| F7 | 09-04-2018 | 08-10-2018 | 5 | 21.0 |
| F8 | 09-04-2018 | 08-10-2018 | 8 | 89.74 |
| F11 | 12-04-2018 | 08-10-2018 | 9 | 68.68 |
| F12 | 12-04-2018 | 24-10-2018 | 12 | 154.11 |
| F13 | 12-04-2018 | 28-10-2018 | 20 | 290.72 |
| F15 | 18-04-2018 | 03-10-2018 | 4 | 49.67 |
| F16 | 01-05-2018 | 24-10-2018 | 10 | 183.30 |
| F17 | 01-05-2018 | 24-10-2018 | 9 | 82.50 |
| F18 | 01-05-2018 | 28-10-2018 | 22 | 221.39 |
| F19 | 02-05-2018 | 24-10-2018 | 7 | 161.94 |
| F20 | 02-05-2018 | 28-10-2018 | 18 | 185.12 |
| F21 | 10-05-2018 | 04-09-2018 | 4 | 38.72 |
| F22 | 10-05-2018 | 24-10-2018 | 14 | 121.38 |

*Appendix 1—table 1 Continued on next page*

*Appendix 1—table 1 Continued*

**Colony-maturation experiments**

| Nest-id | Start date | End date | Max. population | Max. area (cm²) |
|---------|-----------|----------|-----------------|-----------------|
| F23 | 23-05-2018 | 08-10-2018 | 4 | 36.11 |
| F25 | 31-05-2018 | 28-10-2018 | 17 | 171.02 |
| F26 | 31-05-2018 | 28-10-2018 | 14 | 145.75 |
| F27 | 06-05-2018 | 28-10-2018 | 11 | 167.27 |
| F28 | 06-05-2018 | 24-10-2018 | 9 | 89.78 |

**Appendix 1—table 2.** Details of fixed-demographics experiments.

**Fixed-demographics experiments: young**

| Nest-id | Start date | End date | Max. population | Max. area (cm²) |
|---------|-----------|----------|-----------------|-----------------|
| CF-Y67 | 28-04-2021 | 15-06-2021 | 5 | 25.15 |
| CF-Y68 | 28-04-2021 | 15-06-2021 | 5 | 84.27 |
| F-Y35 | 19-10-2020 | 24-12-2020 | 5 | 42.72 |
| F-Y37 | 19-10-2020 | 24-12-2020 | 5 | 30.56 |
| CF-Y64 | 29-04-2021 | 15-06-2021 | 10 | 56.84 |
| CF-Y65 | 29-04-2021 | 15-06-2021 | 10 | 45.47 |
| F-Y30 | 19-10-2020 | 24-12-2020 | 10 | 85.74 |
| F-Y31 | 19-10-2020 | 24-12-2020 | 10 | 92.96 |
| F-Y32 | 19-10-2020 | 24-12-2020 | 10 | 88.44 |
| F-Y33 | 19-10-2020 | 24-12-2020 | 10 | 66.93 |
| CF-Y43 | 10-03-2021 | 23-05-2021 | 15 | 84.98 |
| CF-Y44 | 08-03-2021 | 15-05-2021 | 15 | 74.92 |
| CF-Y45 | 08-03-2021 | 22-05-2021 | 15 | 174.83 |
| F-Y43 | 05-12-2020 | 31-01-2021 | 15 | 132.16 |
| F-Y44 | 05-12-2020 | 31-01-2021 | 15 | 87.94 |
| F-Y45 | 05-12-2020 | 31-01-2021 | 15 | 96.03 |

**Fixed-demographics experiments: old**

| | | | | |
|---------|-----------|----------|-----------------|-----------------|
| CF-O69 | 28-04-2021 | 15-06-2021 | 5 | 27.82 |
| CF-O70 | 28-04-2021 | 10-06-2021 | 5 | 28.64 |
| CF-O71 | 30-04-2021 | 15-06-2021 | 5 | 36.21 |
| CF-O72 | 30-04-2021 | 15-06-2021 | 5 | 15.74 |
| F-O34 | 19-10-2020 | 24-12-2020 | 5 | 7.65 |
| F-O36 | 19-10-2020 | 24-12-2020 | 5 | 32.86 |
| CF-O60 | 28-04-2021 | 16-06-2021 | 10 | 54.65 |
| CF-O61 | 27-04-2021 | 15-06-2021 | 10 | 18.97 |
| CF-O62 | 25-04-2021 | 16-06-2021 | 10 | 28.34 |
| CF-O63 | 28-04-2021 | 16-06-2021 | 10 | 38.13 |
| F-O27 | 19-10-2020 | 24-12-2020 | 10 | 35.99 |
| F-O28 | 19-10-2020 | 24-12-2020 | 10 | 35.39 |

*Continued on next page*

*Continued*

**Fixed-demographics experiments: old**

| F-O29 | 19-10-2020 | 24-12-2020 | 10 | 70.62 |
|---|---|---|---|---|
| CF-O46 | 08-03-2021 | 16-05-2021 | 15 | 76.30 |
| CF-O47 | 08-03-2021 | 14-05-2021 | 15 | 67.61 |
| CF-O48 | 08-03-2021 | 19-05-2021 | 15 | 77.39 |
| CF-O50 | 08-03-2021 | 20-05-2021 | 15 | 95.01 |
| CF-O54 | 08-03-2021 | 12-05-2021 | 15 | 72.97 |
| F-O40 | 30-11-2020 | 31-01-2021 | 15 | 37.37 |
| F-O41 | 30-11-2020 | 31-01-2021 | 15 | 44.07 |
| F-O42 | 30-11-2020 | 31-01-2021 | 15 | 87.98 |

## Population change in the colony-maturation experiments

We quantified the population growth in colony-maturation experiments. Like the nest area, the population growth among the nest varied highly. In some experiments, only one or two workers eclosed ($n$ = 5), while in others, the colony grew more than 15 workers ($n$ = 12). The colony growth increased gradually in all nests until it saturated at a certain level. On average, the population grew logistically, reaching a plateau at roughly 12 ants after around 100 days. We fitted a classical population growth model to describe the population kinetics:

$$\frac{dN}{dt} = rN \left( 1 - \frac{N}{K} \right) \tag{3}$$

On solving the equation, we get,

$$N(t) = \frac{K}{1 + (K-1)e^{-rt}} \tag{4}$$

We used $K$ = 19.0, the maximal number of ants, and $r$ = −0.032, to describe the population growth. We find that the logistic model can describe the population growth relatively well ($R^2$ = 0.98).

## Age quantification from colony-maturation experiments

During the nest photography session, we meticulously recorded the population of ants inhabiting each nest. By analyzing these population records, we were able to construct a comprehensive age demographics profile for the colony. For instance, consider the population change in the following nest:

**Appendix 1—table 3.** Colony demographics from nest population.

| Day | Ant population | New ants | Demographics |
|---|---|---|---|
| … | … | … | … |
| 10 | 1 | | 10 days |
| 11 | 2 | 1 | 1 and 11 days |
| 12 | 3 | 1 | 1, 2, and 12 days |
| 13 | 3 | 0 | 2, 3, and 13 days |
| … | … | … | … |

On day 10, the population remained constant at 1 ant, with the ant being 10 days old. However, on day 11, the population increased by one, indicating the birth of a new ant. Therefore, on day 11, the newly born ant was 1 day old, while the existing ants were 11 days old. Subsequently, we analyzed the rest of the time points, allowing us to estimate the age distribution of ants in the entire nest.

## Percentage contribution to the excavated area

We quantified the percentage contribution of each histogram bin to the excavated area based on the age-dependent excavation model. Our results show that the area excavated between 5 and 13 cm² accounts for 73.76% of the total excavated area, indicating that most ants dug within this range rather than exhibiting extreme variations. Additionally, the mean excavation area is 7.84 cm², with a standard deviation of 3.44 cm², meaning that most values fall between 4.4 and 11.28 cm², which aligns closely with the 5–13 cm² range. Since the majority of excavation is concentrated within this narrow interval, and the mean is well centered within it, these findings suggest that ants excavated similar amounts rather than forming distinct groups with highly different excavation behaviors.

**Appendix 1—table 4.** Percentage contribution to the excavated area by ants within the histogram bin as obtained from the age-dependent model.

| Interval (cm) | Percentage (%) |
| --- | --- |
| 1–2 | 4.3796 |
| 2–3 | 2.9197 |
| 3–4 | 2.5547 |
| 4–5 | 6.2044 |
| 5–6 | 9.8540 |
| 6–7 | 12.7737 |
| 7–8 | 13.8686 |
| 8–9 | 12.0438 |
| 9–10 | 8.7591 |
| 10–11 | 8.0292 |
| 11–12 | 6.2044 |
| 12–13 | 4.3796 |
| 13–14 | 3.2847 |
| 14–15 | 2.5547 |
| 15–16 | 1.8248 |
| 16–17 | 0.36496 |

