## [Editor Report · eLife Assessment]

This study presents an **important** finding that ant nest structure and digging behavior depend on ant age demographics for a ground-dwelling ant species (Camponotus fellah). By asking whether ants employ age-polyethism in excavation, the authors address a long-standing question about how individuals in collectives determine the overall state of the task they must perform. The experimental evidence that the age of the ants and the group composition affect the digging of tunnels is **convincing**, and their model is able to replicate the colony's excavation dynamics qualitatively, results that may prove to be a key consideration for interpreting results from other studies in the field of social insect behavior.

---

## [Referee Report · Reviewer #1 (Public review)]

This study investigates how ant group demographics influence nest structures and group behaviors of Camponotus fellah ants, a ground-dwelling carpenter ant species (found locally in Israel) that build subterranean nest structures. Using a quasi-2D cell filled with artificial sand, the authors perform two complementary sets of experiments to try to link group behavior and nest structure: first, the authors place a mated queen and several pupae into their cell and observe the structures that emerge both before and after the pupae eclose (i.e., "colony maturation" experiments); second, the authors create small groups (of 5, 10, or 15 ants, each including a queen) within a narrow age range (i.e., "fixed demographic" experiments) to explore the dependence of age on construction. Some of the fixed demographic instantiations included a manually induced catastrophic collapse event; the authors then compared emergency repair behavior to natural nest creation. Finally, the authors introduce a modified logistic growth model to describe the time-dependent nest area. The modification introduced parameters that allow for age-dependent behavior, and the authors use their fixed demographic experiments to set these parameters, and then apply the model to interpret the behavior of the colony maturation experiments. The main results of this paper are that for natural nest construction, nest areas, and morphologies depend on the age demographics of ants in the experiments: younger ants create larger nests and angled tunnels, while older ants tend to dig less and build predominantly vertical tunnels; in contrast, emergency response seems to elicit digging in ants of all ages to repair the nest.

The experimental results are convincing, providing new information and important insights into nest and colony growth in a social insect species. A model, inspired by previous work but modified to capture experimental results, is in reasonable agreement with experiments and is more biologically relevant than previous models.

---

## [Referee Report · Reviewer #2 (Public review)]

I enjoyed this paper and its examination of the relationship between overall density and age polyethism to reduce the computational complexity required to match nest size with population. I had some questions about the requirement that growth is infinite in such a solution, but these have been addressed by the authors in the responses and updated manuscript. I also enjoyed the discussion of whether collective behaviour is an appropriate framework in systems in which agents (or individuals) differ in the behavioural rules they employ, according to age, location, or information state. This is especially important in a system like social insects, typically held as a classic example of individual-as-subservient to whole, and therefore most likely to employ universal rules of behaviour. The current paper demonstrates a potentially continuous age-related change in target behaviour (excavation), and suggests an elegant and minimal solution to the requirement for building according to need in ants, avoiding the invocation of potentially complex cognitive mechanisms, or information states that all individuals must have access to in order to have an adaptive excavation output.

The authors have addressed questions I had in the review process and the manuscripts is now clear in its communication and conclusions.

The modelling approach is compelling, also allowing extrapolation to other group sizes and even other species. This to me is the main strength of the paper, as the answer to the question of whether it is younger or older ants that primarily excavate nests could have been answered by an individual tracking approach (albeit there are practical limitations to this, especially in the observation nest setup, as the authors point out). The analysis of the tunnel structure is also an important piece of the puzzle, and I really like the overall study.

---

## [Author Response]

The following is the authors’ response to the previous reviews.

**Reviewer #1 (Public review):**
This study investigates how ant group demographics influence nest structures and group behaviors of Camponotus fellah ants, a ground-dwelling carpenter ant species (found locally in Israel) that build subterranean nest structures. Using a quasi-2D cell filled with artificial sand, the authors perform two complementary sets of experiments to try to link group behavior and nest structure: first, the authors place a mated queen and several pupae into their cell and observe the structures that emerge both before and after the pupae eclose (i.e., "colony maturation" experiments); second, the authors create small groups (of 5,10, or 15 ants, each including a queen) within a narrow age range (i.e., "fixed demographic" experiments) to explore the dependence of age on construction. Some of the fixed demographic instantiations included a manually induced catastrophic collapse event; the authors then compared emergency repair behavior to natural nest creation. Finally, the authors introduce a modified logistic growth model to describe the time-dependent nest area. The modification introduced parameters that allow for age-dependent behavior, and the authors use their fixed demographic experiments to set these parameters, and then apply the model to interpret the behavior of the colony maturation experiments. The main results of this paper are that for natural nest construction, nest areas, and morphologies depend on the age demographics of ants in the experiments: younger ants create larger nests and angled tunnels, while older ants tend to dig less and build predominantly vertical tunnels; in contrast, emergency response seems to elicit digging in ants of all ages to repair the nest.The experimental results are solid, providing new information and important insights into nest and colony growth in a social insect species. As presented, I still have some reservations about the model's contribution to a deeper understanding of the system. Additional context and explanation of the model, implications, and limitations would be helpful for readers.

We sincerely thank Reviewer #1 for the time and effort dedicated to our manuscript's detailed review and assessment. The new revision suggestions were constructive, and we have provided a point-by-point response to address them.

**Reviewer #2 (Public review):**
I enjoyed this paper and its examination of the relationship between overall density and age polyethism to reduce the computational complexity required to match nest size with population. I had some questions about the requirement that growth is infinite in such a solution, but these have been addressed by the authors in the responses and the updated manuscript. I also enjoyed the discussion of whether collective behaviour is an appropriate framework in systems in which agents (or individuals) differ in the behavioural rules they employ, according to age, location, or information state. This is especially important in a system like social insects, typically held as a classic example of individual-as-subservient to whole, and therefore most likely to employ universal rules of behaviour. The current paper demonstrates a potentially continuous age-related change in target behaviour (excavation), and suggests an elegant and minimal solution to the requirement for building according to need in ants, avoiding the invocation of potentially complex cognitive mechanisms, or information states that all individuals must have access to in order to have an adaptive excavation output.The authors have addressed questions I had in the review process and the manuscript is now clear in its communication and conclusions.The modelling approach is compelling, also allowing extrapolation to other group sizes and even other species. This to me is the main strength of the paper, as the answer to the question of whether it is younger or older ants that primarily excavate nests could have been answered by an individual tracking approach (albeit there are practical limitations to this, especially in the observation nest setup, as the authors point out). The analysis of the tunnel structure is also an important piece of the puzzle, and I really like the overall study.

We sincerely thank Reviewer #2 for the time and effort dedicated to our manuscript's detailed review and assessment.

**Reviewer #1 (Recommendations for the authors):**
Thank you for the modifications. I found much of the additional information very helpful. I do still have a few comments, which I will include below.

We thank the reviewer for this comment

The authors provide some additional citations for the model, however, the ODE in refs 24 and 30 is different from what the authors present here, and different from what is presented in ref 29. Specifically, the additional "volume" term that multiplies the entire equation. Can the authors provide some additional context for their model in comparison to these models as well as how their model relates to other work?

We thank the reviewer for this question. The primary difference between the logistic model (reference number: 24,30), and the saturation model (reference number: 29) is rooted in their assumptions on the scaling of the active number of ants that participate in the nest excavation and the nest volume.

The logistic growth model (𝑑𝑉/𝑑𝑡 = α𝑉(1-V/Vs)) describes the excavation in fixed-sized colonies (50, 100, 200) through a balance of two key processes : (1) positive feedback (α𝑉), where the digging efficiency increases with the nest size, and (2) negative feedback (1-V/Vs), where growth slows as the nest approaches a saturation (Vs). The model assumes that the number of actively excavating ants is linearly proportional to the nest volume (V). This represents a scenario where a large nest contains or can support more workers, which in turn increases the digging rates. While this does not require explicit communication between individuals, ants indirectly sense the global nest volume through stigmergic cues, such as pheromone depositions, encounter rates, while ignoring individual differences in age.

In contrast, the saturation model (𝑑𝑉/𝑑𝑡 = α𝑉(1-V/Vs)) assumes a constant number of ants is working throughout the excavation. The digging rate is therefore independent of the nest volume, this model imposes a different cognitive requirement ants must somehow assess the global nest slowing only due to the saturation term (1-V/Vs) as the nest approaches its target size. However, volume (V) and the overall number of ants in the nest. Thus, rather than relying on local cues, ants need more explicit communication or a sophisticated global perception mechanism that allows ants to sense the nest volume and the nest population to adjust the digging rates accordingly. Therefore, this model requires a more complex and less biologically plausible mechanism than the logistic model.

In our age-dependent digging model in the manuscript, we explicitly sum the contribution of each ant towards the nest area expansion based on its age-dependent digging threshold (quantified from fixed demographics experiments) the sum over Thus, the term ‘V’ in the ‘ 𝑉(1-V/Vs) takes the same effect as sum over all ants in the equation (2) of our manuscript; they describe how the total excavation rate scales with the number of individuals. Under the simplifying assumption that the number of ants is proportional to the nest volume ‘V’, and that all ants dig at a constant rate, our equation (2) in the manuscript reduces to the logistic equation ‘𝑉(1-V/Vs)’ This implies that each ant individually assesses the nest volume and then digs at a rate ‘(1-V/Vs)’.

Thus, we adopted the simpler model from the previously published ones, in which ants individually react to the local density cues and regulate their digging. This approach does not require a global assessment of the nest volume or the number of ants; a local perception of density triggers each ant’s decision to dig, likely modulated by the frequency of social contacts or chemical concentration, which serves as an indicator of the global nest area. The ant compares this locally perceived density to an innate, age-specific threshold. If the perceived local density exceeds its threshold (indicating insufficient area), it digs; otherwise, there is no digging. Thus, excavation dynamics in maturing colonies emerge from this collective response to local density cues, without any individual need to directly assess the global nest volume (V) or having explicit knowledge of the colony size (N).

As suggested by the reviewer, we have added these points to the discussion, contrasting the previously published models with our age-dependent excavation models **(line numbers: 283-290)** “In our study, we adopted the simpler version of previously published age-independent excavation models, where individuals respond to local stigmergic cues such as encounter rates or pheromone concentrations, which serve as a proxy for the global nest volume (24,30). We minimally modified this model to include age-dependent density targets. According to our age-dependent digging model, each ant compares this perceived local density to its own innate age-specific digging threshold as quantified from the fixed demographics experiments. If the perceived local density exceeds its age-dependent area threshold (indicating insufficient area), it digs; otherwise, there is no digging. This mechanism eliminates the need for cognitively demanding global assessment of the total nest volume or the overall colony population, a requirement for the saturation model (29)”.

I still find it a little concerning that the age-independent model, though it cannot be correct, fits the data better than the age-dependent modification. It seems to me the models presented in refs 24, 29, and 30, which served as inspiration for the one presented here, do not have any deep theoretical origin, but were chosen for "being consistent with" the observed overall excavated volumes. Is this correct, and if so, how much can/should be gleaned about behavior from these models? Please provide some discussion of what is reasonable to expect from such a model as well as what the limitations might be.

We thank the reviewer for the comment.

In our study, we make an important assumption, as described in the lines (line number : 161 - 164) of the manuscript, that ants rely on local cues during nest excavation, and individuals cannot distinguish between the fixed demographics and colony maturation conditions. This implies that the age-dependent target area identified in the fixed demographics experiments should also account for the excavation dynamics seen in the colony maturation experiments.

From the fixed demographics young and old experiments, we directly quantified that the younger ants excavate a significantly larger area than the older ants for the same group size. This age-dependent digging propensity is an experimental result, and not a model output.

We agree that the age-independent model fits the colony maturation experiments well, even though it's not a statistically better fit than the age-dependent model. However, the age-independent models in the references (24,29,30) fail to explain the empirically obtained excavation dynamics in the fixed demographics, young and old colonies. If indeed these models were true, then we would have observed similar excavated areas between the colony maturation, fixed demographics, young, and older colonies of the same size. Thus, the inconsistency of these models confirms that age-independent assumptions are biologically inadequate. These details are explicitly mentioned in lines (304 - 309).

We believe that our model’s value is in providing a plausible explanation for the observed excavation dynamics in the colony maturation experiments, and generating testable predictions (Figure 4. C, and 4.D, described in lines 199 - 216) about the percentage contribution of different age cohorts and queens to the excavated area from the colony maturation experiments. This prediction would not be possible with an age-independent model.

Minor comments:Figure 2A: Please use a color other than white for the model... this curve is still very hard to see

We thank the reviewer for the comment. The colour is changed to yellow.

Figure 4A: Should quoted confidence intervals for slope and intercept be swapped?

Yes, we thank the reviewer for pointing this out. The labels for the slope and intercept were swapped. We corrected this in the current revised version 2.

Figure 5 D-F: Can the authors show data points and confidence intervals instead of bar graphs? The error bars dipping below zero do not clearly represent the data.

We thank the reviewer for the comment. We now show the individual data points from each treatment with the 95% Confidence Interval of the mean.